

# Thermodynamic and hydrological drivers of the subsurface thermal regime in Central Spain

Félix García-Pereira[1], Jesús Fidel González-Rouco[1], Thomas Schmid[2], Camilo Melo-Aguilar[3], Cristina Vegas-Cañas[1], Norman Julius Steinert[4], Pedro José Roldán-Gómez[1], Francisco José Cuesta-Valero[5], Almudena García-García[5], Hugo Beltrami[6], and Philipp de Vrese[7]

[1]Complutense University of Madrid, Faculty of Physical Sciences, and Geosciences Institute (UCM-CSIC), Madrid, Spain
[2]Department of Environment, CIEMAT, Madrid, Spain
[3]Balearic Ocean Centre, Spanish Institute of Oceanography (IEO-CSIC), Palma de Mallorca, Spain
[4]NORCE, Norwegian Research Centre, Climate and Environment, Bergen, Norway
[5]Helmholtz-Centre for Environmental Research (UFZ), Leipzig, Germany
[6]Climate and Atmospheric Sciences Institute, St. Francis Xavier University, Antigonish, Canada
[7]Max Planck Institute for Meteorology, Hamburg, Germany

**Correspondence:** Félix García-Pereira (felgar03@ucm.es), Cristina Vegas-Cañas (cvegas@ucm.es)

**Abstract.** An assessment of the soil and bedrock thermal structure of the Sierra de Guadarrama, in Central Spain, is provided using subsurface and ground surface temperature data coming from four deep (20 m) monitoring profiles belonging to the Guadarrama Monitoring Network (GuMNet), and two shallow (1 m) from the Spanish Meteorology Service (AEMET), covering the timespan of 2015-2021 and 1989-2018, respectively. An evaluation of air and ground surface temperature coupling

shows soil insulation due to snow cover is the main source of seasonal decoupling, being especially relevant in winter at high altitude sites. Temperature propagation in the subsurface is characterised by assuming a heat conductive regime, by considering apparent thermal diffusivity values derived from the amplitude attenuation and phase shift of the annual cycle with depth. For the deep profiles, the apparent thermal diffusivity ranges from 1 to 1.3 $10^{-6}$ $m^2s^{-1}$, consistent with values for gneiss and granite, the major bedrock components in the Sierra de Guadarrama. However, thermal diffusivity is lower and more heterogeneous

in the soil layers close to the surface (0.4-0.8 $10^{-6}$ $m^2s^{-1}$). An increase of diffusivity with depth is observed, being generally larger in the soil-bedrock transition, at 4-8 m depth. A new method based on the spectral attenuation of temperature harmonics allows for analysing thermal diffusivity from high frequency changes in the soil near the surface at short timescales. The results are relevant for the understanding of soil thermodynamics in relation to other soil properties and suggest that changes in heat diffusivity are related to soil moisture content changes, which makes this method a potential tool in soil drought and water

resource availability reconstruction from soil temperature data.

## 1  Introduction

The last decades have witnessed substantial steps forward in knowledge of the climate system and the ongoing anthropogenic climate change (Chen et al., 2021). An increase in observational datasets (Hansen et al., 2010; Lawrimore et al., 2011; Jones et al., 2012; Rohde et al., 2013; Lenssen et al., 2019; Osborn et al., 2021), and progressively more realistic climate model



ensembles (von Storch, 2010; Taylor et al., 2012; Eyring et al., 2016; Maher et al., 2019) has allowed for attributing a global mean near-surface air temperature rise of more than $1^oC$ since the end of the 19th century to human activities (Chen et al., 2021; Gulev et al., 2021). At the global scale, the increase of air temperature affects the ocean and land surface disproportionally. Due to their differences in moisture availability for evaporative cooling, the land warms faster than the ocean (Sutton et al., 2007). It is therefore vital to understand the interaction between air and land temperatures and its propagation into the soil and
subsoil, which has been shown to be important for land surface processes and meteorological extreme events (Rahmstorf and Coumou, 2011; Seneviratne et al., 2021).

Ground surface temperature (GST) and surface air temperature (SAT) are coupled at long-term scales (Melo-Aguilar et al., 2018), and GST perturbations are subsequently propagated into depth by heat conduction (Carslaw and Jaeger, 1959), hence SAT changes are transmitted into the subsurface. A warming of near-surface soil has an impact on its biogeochemical activity
(Soong et al., 2021), enhancing metabolic activity of microorganisms, mineral weathering, and decomposition of soil organic matter (Schlesinger and Emily, 2013). It also affects soil hydrology (Krakauer et al., 2013) and increases soil respiration (Pries et al., 2017a), contributing to an acceleration of the terrestrial carbon cycle (Bond-Lamberty and Thomson, 2010; Pries et al., 2017b). Moreover, soil warming has a relevant influence on permafrost thawing, which can potentially contribute to the release of massive amounts of carbon to the atmosphere (Turetsky et al., 2019), contamination problems derived from mercury release
to the atmosphere and soil waters (Schaefer et al., 2020), or even permafrost collapse (Abbott and Jones, 2015; Pegoraro et al., 2021), with implications for the climate variability of high latitude regions (Andresen et al., 2020; Burke et al., 2020) and potentially for global climate (de Vrese et al., 2022). The subsurface is not only sensitive to present global warming, but also to past climate changes. Deep subsurface temperatures permit to recover past SAT histories by Borehole Temperature Profile (BTP) inversion techniques (Mareschal and Beltrami, 1992; Huang et al., 2000; Pollack and Smerdon, 2004; Jaume-
Santero et al., 2016; Cuesta-Valero et al., 2022a), which make them a valuable source for climate reconstruction as far as the two fundamental hypothesis are sustained: SAT and GST changes are coupled at long-term scales and GST changes are propagated downwards by heat conduction. Therefore, assessing SAT-GST coupling and heat conduction in the subsurface from observational sources becomes crucial to understand its role in past evolution to present climate conditions.

Support for SAT-GST coupling has been found so far, both from observational temperature data at some sites in North
America (e.g., Beltrami and Kellman, 2003; Bartlett et al., 2005, 2006) and Europe (e.g., Cermak et al., 2014; Cermak and Bodri, 2016; Melo-Aguilar et al., 2022; Petersen, 2022; Dorau et al., 2022), and climate simulations (González-Rouco et al., 2006; García-García et al., 2016; Melo-Aguilar et al., 2018; García-García et al., 2019; González-Rouco et al., 2021). The results show a strong coupling at long-term scales globally, with some degree of seasonal decoupling due to snow cover (Beltrami and Kellman, 2003) or changes in the surface energy balance (Smerdon et al., 2004; Melo-Aguilar et al., 2018) at
regional scales. Examining whether conduction is the main heat transport mechanism in the subsurface has been usually tackled by assessing the propagation of temperature perturbations and verifying that they experience an amplitude damping and phase shift with depth to an extent controlled by thermal diffusivity, a parameter that governs heat conduction processes (Carslaw and Jaeger, 1959). In practice, this assessment can be easily simplified to studying the propagation of a single frequency harmonic (Hurley and Wiltshire, 1993; Smerdon et al., 2003). Previous works have targeted the annual wave with depth since it is the





most stable harmonic that can be fitted within the range of GST timescales. Nevertheless, this approach requires a continuous
       monitorization of subsurface temperature during a relatively long lapse of time, of at least a few years. That has only been
       possible at a scarce number of places in North America (Smerdon et al., 2004; Bartlett et al., 2006) and Europe (Smerdon et al.,
       2004; Demetrescu et al., 2007; Dědeček et al., 2013; Melo-Aguilar et al., 2022). Moreover, the analysis might be hampered
       by potential deviations from the conductive regime in the soil near the surface due to biological, chemical, and hydrological
processes (Gao et al., 2008; Tong et al., 2017). A deeper insight into the soil thermal structure is required to understand
       changes in its thermal properties with depth, relevant for Land Surface Models (LSMs) and therefore with implications on
       land-atmosphere interactions affecting Earth System (Smerdon and Stieglitz, 2006; Ekici et al., 2014; Hermoso de Mendoza
       et al., 2020; Steinert et al., 2021) and forecast models (Miralles et al., 2019). A better characterization of subsurface thermal
       properties can also contribute to derive more accurate land heat uptake estimates, which is the second component contributing
the most to terrestrial energy partitioning (Cuesta-Valero et al., 2022b; von Schuckmann et al., 2022).

       This paper tests the compliance of the SAT-GST coupling and conductive thermal propagation hypotheses in the penetration
       of SAT perturbations into the subsurface at a high-altitude area in Central Spain. This is achieved by determining the apparent
       soil thermal diffusivity from four short and two relatively long subsurface temperature records obtained at six sites in the area
       of the Sierra de Guadarrama, and studying its variability in time and depth through the soil and down to the first meters of
bedrock. The complex orography of the Sierra de Guadarrama has been shown to dominate the spatial mean and variability
       of SAT, while temporal variability has been found to be very homogeneous over the region (Vegas-Cañas et al., 2020). In a
       first stage, we explore the SAT-GST coupling at different sites, considering the influence of processes at seasonal timescales.
       Then, the propagation of the annual wave with depth is studied. This permits to test the conductive hypothesis and establish
       a comparison between different sites by using the standard approach of focusing on the propagation of the annual wave with
depth. The analysis is further extended by assessing changes in thermal diffusivity in the subsurface temperature profile, which
       are linked to the soil-bedrock mineral composition transition observed from samples extracted at each of the sites.

       The study of the conductive regime also explores changes in apparent thermal diffusivity with time. This is feasible for
       the longer records for which the analysis can be performed focusing on different multi-annual long temporal intervals. This
       approach is however hampered by the short time span of observations at most sites considered herein. This problem is overcome
by introducing a new method that makes use of the information in the spectral domain. Instead of considering only the annual
       cycle, the propagation of the complete range of harmonics of different frequencies that contribute to temperature variability is
       used. This allows for focusing on intra-annual timescales and using shorter time series to estimate soil thermal diffusivity near
       the surface. The fact that apparent thermal diffusivity values can be retrieved from shorter timespans also permits to assess its
       changes through time and potential links to changes in soil moisture content (Sorour et al., 1990; Fuchs et al., 2021).

The manuscript is structured as follows. In Section 2, the data and their quality control are described. Section 3 explains the
       methodologies used. In Section 4 results of the analysis are presented and these are discussed in Section 5.



## 2 Data

The subsurface data analysed hereafter consist of temperature time series from six locations distributed over the Sierra de Guadarrama (see Table 1). The Sierra de Guadarrama is part of the Sistema Central, a mountain system which splits the Spanish Central Plateau into a southeastern side, with altitudes of around 600 m above sea level (m.a.s.l.), and a northwestern side, of higher elevation (ca. 750 m.a.s.l.). Elevations in the Sierra span from 900 at the foothills to ca. 2200-2400 m.a.s.l. at the summits. Two sites, Puerto de Navacerrada and Segovia, belong to the Spanish Meteorological Service (Agencia Estatal de Meteorología, AEMET), and the rest are part of the Guadarrama Monitoring Network (GuMNet). The latter was created in 2014 with the aim of gaining further insights into climate variability in mountain environments in Central Spain (Vegas-Cañas et al., 2020). GuMNet consists of 10 atmosphere and subsurface monitoring stations, which cover a vertical gradient ranging from 900 to 2200 m.a.s.l., spreading over the mountainous terrain of the Sierra de Guadarrama. In this work, four stations were selected that provide information of both atmospheric variables and of subsurface temperatures (Fig. 1a).

The subsurface thermal regime is monitored by two different arrangements. Near-surface soil temperatures are measured at various levels in trenches (TRCH; Fig. 1b) at all sites. Trenches are excavations forming a slope in which the frontal wall was used both to analyse the vertical structure of soil horizons (Brady and Weil, 2017) and to insert temperature sensors down to a depth of about 1 m. Trenches were refilled with the previously extracted material. Nevertheless, this solution is limited to the uppermost layers, and deeper temperatures in the soil and bedrock are monitored within cased boreholes. They consist of a small diameter (ca. 76 mm) cylindrical drill in depth where a casing was inserted and afterwards filled up with a silicone gel. Temperatures therein are measured with calibrated thermistors embedded inside the casing (BRH; Fig. 1c,d). For the sake of having finer vertical resolution in the soil near the surface and some level of redundancy, two boreholes were installed at these sites: a deep one going down to 20 m (BRH 20 m in Fig. 1 and Table 2), and a shallow one of 2 m (BRH 2 m in Fig. 1). Table 2 provides a detailed list of subsurface temperature levels with available observations at every site.

During the borehole drilling process at the GuMNet sites, the extracted cores were preserved. They were subsequently sampled and characterised. The result is shown in Fig. 2, where the limits between the soil, sediment and rock layers were determined. The superficial soil layer is made up of different soil horizons (Brady and Weil, 2017), containing an O and/or A (i.e. topsoil) followed by E and/or B (i.e. subsoil) and several C (i.e. parent material) horizons that reach a depth of approximately 1 m. The underlying sediment layer contains eroded material of loose debris rocks that have accumulated at the corresponding locations. The CTS, HYS, and RPI profiles are located over gneiss rock and HRR on granite rock. In all profiles, the upper layers have weathered bedrock and with depth this bedrock is unweathered and compact. This information is used in Section 4 to compare changes in apparent thermal diffusivity to changes in mineral composition and textures with depth.

Subsurface temperature measurements were taken at 10-minute time resolution at every GuMNet station. As a first step prior to the analysis carried out in this work, subsurface data are subjected to quality control and resampling procedures. AEMET stations (NVC and SGV) were independently processed and resampled to daily resolution, as described in Melo-Aguilar et al. (2022). The quality control of GuMNet sites focused first on removing data outliers. Since subsurface temperature sensors are connected in series both in boreholes and trenches (see Fig. 1b-d), most often erroneous extreme values are recorded at all levels



simultaneously. Thus, to prevent discarding values that might be correct meteorological extremes, this correction is applied both for boreholes and trenches, using the deepest time series to detect the erroneous values. The lowermost level is used because it shows the lowest temperature variability due to conductive damping of the surface signal. This reduces the range of high-frequency variability in the series except for the erroneous data, which makes non-reliable outliers easier to be detected (Fig. 3).

First, outliers flagged as nonphysically plausible are removed, i.e. values falling below 230 K or exceeding 350 K. The detection of outliers below 230 K can be considered redundant, since the next step would have detected all the values lying beneath these threshold. Second, the remaining spikes are flagged and discarded. Spikes are detected by calculating the sequential differences of each value minus its preceeding one and screening the values in the interval $(Q_1 - 3IQR, Q_3 + 3IQR)$, where $Q_1$, $Q_3$, and $IQR$ are the first and third quartiles, and the interquartile range of the distribution of temperature differences (Tukey, 1977).

Time steps whose difference values are missing or outside of the screening interval are removed. Fig. 3 shows an example of such spike screening for the BRH 2 m at CTS. The outliers depict coinciding failures at all levels. The procedure is very conservative as the bottom levels bear very low temperature variability and all spikes are identified and removed at each level. Note the decrease in variability with depth away from the subsurface, which is in agreement with conductive law (Carslaw and Jaeger, 1959). Once outliers have been removed, time series are subjected to a one-by-one inspection, and periods with clear

drifts or strong changes in variability are manually removed (check Table 3 for a list of removed periods at CTS and HYS). Fig. 3 also shows an example of one of such periods for the 1.5 m depth temperatures at CTS (green shading).

After applying the aforementioned quality control procedures, the GuMNet series are resampled at hourly resolution, which filters out noisy intra-hourly variability unnecessary for the subsequent analysis of the data. This is attained by calculating pseudo-hourly time series, which is carried out with a Nearest Neighbour Resampling technique (Brandsma and Können, 2006)

by selecting the 10-minute data closest to the hour. Finally, the resulting series are averaged to obtain daily mean subsurface temperatures that are subsequently used to analyse the propagation of the annual cycle.

In addition to subsurface temperature data, all the stations include SAT measurements as well as soil moisture content (SH) within the trench monitoring levels. Snow cover data are also available at CTS, HYS, and RPI (see codes in Table 1). These additional data are used to explore potential drivers of the GST-SAT seasonal decoupling and temporal changes in soil apparent

thermal diffusivity at some sites.

## 3  Methodology

The SAT-GST seasonal coupling is explored by quantifying the offset of seasonal (December-January-February, DJF; June-July-August, JJA) and annual 2 m air and BRH 2 m top level mean temperatures. This assessment is extended in time by making use of a correlation analysis on seasonal and annual temperatures.

Furthermore, the subsurface temperature distribution is analysed by assuming that GST perturbations are propagated downwards by heat conduction. Considering that every level is at a thermal equilibrum state, horizontal heat transport may be neglected. Thus, the problem is reduced to resolve the one-dimensional time-dependent heat conduction equation, $\partial T/\partial t = \alpha \partial^2 T/\partial z^2$. If GST is assumed to follow a sinusoidal (e.g. annual) cycle, subsurface temperatures at any depth $z$ can be derived



as follows (Hurley and Wiltshire, 1993):

$$T(z,t) = T_0 + A_0 e^{-z\sqrt{\pi f_a/\alpha}} \cos(2\pi f_a t - z\sqrt{\pi f_a/\alpha}), \tag{1}$$

where $T_0$ and $A_0$ are the mean temperature and wave amplitude of the annual cycle at the ground surface, respectively, $f_a$ is the frequency of the annual cycle, and $\alpha$ is the thermal diffusivity, which is a direct function of thermal conductivity ($k$), and inverse of density ($\rho$) and heat capacity ($c$), i.e. $\alpha = k/\rho c$. Equation 1 shows that the annual cycle amplitude is attenuated and phase shifted with depth. The amplitude attenuation (phase shift) is exponential (linear) with depth and dependent on the
thermal diffusivity, i.e. increasing this parameter produces a greater amplitude attenuation (phase shift).

Amplitude and phase values of the annual cycle at every level are obtained by least-squares fitting subsurface temperature series at a daily resolution to an annual-period sinusoidal wave. Then, a linear regression analysis on the annual cycle amplitude (phase shift) values with depth yields estimates of the apparent thermal diffusivity of the subsurface, since the resulting slope is equal to $\sqrt{\pi f_a/\alpha}$ in Equation 1. Amplitude (phase shift) values coming from every installation (TRCH 1 m, BRH 2 m, and
BRH 20 m) are independently adjusted and normalised to prevent any disruption due to blending data from different sources when assessing heat propagation from ground surface to 20 m depth. Normalization was attained by dividing (substracting) amplitude (phase shift) of the annual cycle at every level by the amplitude (phase shift) value at the ground surface yielded by the linear regression adjustment. Once normalised, all amplitude (phase shift) values are brought together and linearly adjusted to obtain a single estimate of thermal diffusivity for the whole subsurface profile at every station. A similar methodological
approach based on the analysis of the annual cycle has been widely used in the literature (e.g., Smerdon et al., 2004; Pollack et al., 2005; González-Rouco et al., 2009) and it will be referred to as the classic analytical approach, CA hereafter. This CA frame will also be used to estimate changes in apparent thermal diffusivity with depth, either considering changes between pairs of subsurface levels or using a two-phase regression analysis (Solow, 1987, 1995; Melo-Aguilar et al., 2018) to identify depths where significant changes in apparent thermal diffusivity occur.
Beyond the regular nature of the annual cycle, GST variability encloses perturbations at different frequencies. If GST is considered as a sum of temperature harmonics, each of those would be also propagated with depth following Carslaw and Jaeger (1959):

$$T(z,t) = T_0 + \sum_{i=1}^{N} A_0 e^{-z\sqrt{\pi f_i/\alpha}} \cos(2\pi f_i - z\sqrt{\pi f_i/\alpha}), \tag{2}$$

where $f_i$ is the frequency of each wave component in a spectral decomposition, as yielded by a Fourier periodogram for
instance (Brigham and Morrow, 1967). Equation 2 shows amplitude attenuation and phase shift are frequency-dependent. More precisely, amplitude attenuation grows exponentially with frequency, i.e. high-frequency harmonics decay increasingly faster with depth, whilst low-frequency perturbations can propagate deeper. For instance, the daily cycle hardly reaches a depth of 50 cm, while the annual cycle propagates down to 15 m (Putnam and Chapman, 1996). Hence, the subsurface acts as a lowpass filter. The temperature spectral attenuation with depth can be expressed as:



$\quad \zeta = \sqrt{P_z(\omega)/P_{GST}(\omega)} = e^{-z\sqrt{\pi f/\alpha}}$  $\qquad$ (3)

where $P_{GST}(\omega)$ and $P_z(\omega)$ are the spectral power densities of GST and that at a given level $z$, respectively:

$$P_z(\omega) = \frac{1}{N_s}\left|\sum_{t=1}^{N} T(z,t)e^{-i\omega t}\right|^2, \; with \; \omega = 2\pi f \qquad (4)$$

where $N_s$ is the temporal resolution of the temperature time signal at $z$, $T(z,t)$, $f$ is the frequency and $\omega$ is the angular

frequency of the different wave components of $T(z,t)$. In order to minimise the undesirable effect of Gibbs' noise at high

frequencies, $P$ is calculated making use of Welch's modified periodogram (Stoica and Moses, 1997). Since periodogram

calculation requires evenly spaced data, data gaps were previously filled by linear interpolation. As the spectral attenuation

must follow an exponential relationship with frequency, the spectral amplitude attenuation curves attained from the analysis

of Equation 3 are subsequently linearised (i.e. taking the natural logarithm) and linearly fitted by least-squares to the values

obtained at all available depths:

$\quad (ln\zeta)^2 = \frac{\pi z^2}{\alpha} f$  $\qquad$ (5)

Estimates for the apparent thermal diffusivity of the soil layer in between the ground surface and level $z$ can be retrieved by

dividing $\pi z^2$ by the regression line slope, $\pi z^2/\alpha$, in Equation 5. To prevent that poor signal-to-noise ratios at high-frequencies

bias the analysis, amplitude attenuation values at frequencies higher than the corresponding $e^2$-fold decay cut-off frequency

(i.e. spectral harmonics attenuated more than $e^2$ times) are filtered out at every level. To determine this frequency at every

site and level, near-surface soil thermal diffusivity values coming from the CA are used. This alternative approach to the CA

will from now on be called the spectral method (SM). Whereas the CA needs at least a few years of data to provide reliable

fits of the annual cycle to subsequently retrieve the apparent thermal diffusivity, the SM can be applied to data at shorter time

intervals. That makes it suitable to analyse short time series, which is the case of some of the time series presented in this work.

Consequently, the SM is used in this work to retrieve apparent thermal diffusivity values in the first meter of the soil at every

station. These values are then compared to the estimates coming from the CA to assess the robustness of the results from both

methods. Furthermore, the SM allows to gain some insight into short-term changes of soil apparent thermal diffusivity and its

potential relation with changes in snow cover and soil moisture content.

## 4  Results

### 4.1  Temperature variability at the ground surface

In order to illustrate the amplitude attenuation and shift with depth, Fig. 4 compiles all the subsurface temperature series at

hourly resolution coming from BRH 20 m, BRH 2 m, and TRCH 1 m at HRR; SAT is also represented to illustrate the coupling





at the surface. A first inspection reveals that the only deterministic temperature variation at all levels is the annual cycle. The daily cycle, while regular in occurrence, changes a lot in magnitude due to weather conditions and seasonality. The annual cycle is the most prominent at the ground surface (ca. 9 K amplitude) despite that both SAT and GST also show remarkable daily temperature variations (ca. 4 K amplitude). It is notorious how the annual cycle decreases in amplitude with depth to be imperceptible at 15 m, where annual temperature variations are of the same order of the accuracy of the sensor, i.e. 0.1 K. Further, the increasing lag in the occurrence of temperature maxima (minima) in summer (winter) is indicative of the phase shift due to conductive propagation of the temperature variations. Both phenomena can be inferred either by observing the subsurface temperature series or the sinusoidal curves adjusted to an annual cycle (blue dashed lines). Further, SAT and GST daily variabilities are similar along the year except in winter, when GST variability is smaller in comparison with SAT. For the rest of soil levels, warm or cold anomalies that last a few weeks are propagated only in the first meter of the soil.

The long-term mean temperatures with depth for CTS, HYS, RPI, and HRR are shown in Fig. 5. The annual vertical mean temperature of the profile, which is observed below the depth at which the annual cycle has been attenuated (ca. 15 m), is remarkably higher at HRR (286.8 K) than at CTS (281.3 K), HYS (279.2 K), and RPI (280.9 K). The JJA (DJF) temperature profile further depicts the seasonal temperature variations towards warmer (colder) values in summer (winter), and the amplitude attenuation of these seasonal variations with depth. JJA and DJF temperature profiles converge at a depth of around 5 m at all sites to then cross over, which is a sign of the phase shift of the annual cycle. With regard to SAT-GST coupling, Fig. 5 shows the annual cycle is slightly weaker for SAT than GST due to radiative warming (cooling) of the ground surface, being SAT-GST anomalies negative (positive) in summer (winter). The distribution of SAT and GST mean values of the sites agree with their different altitudes, which yields a vertical gradient of -5.81 K km$^{-1}$ in SAT given by Vegas-Cañas et al. (2020) for the area of the Sierra de Guadarrama. SATs in Fig. 5 comply well with this value, rendering a gradient of -6.55 K km$^{-1}$, larger in magnitude due to this subset of sites being distributed over the higher altitudes of the region. Subsurface temperatures in depth follow the distribution of SAT, with colder (warmer) temperatures at the higher (lower) sites, except for the case of RPI and CTS. Although RPI shows lower temperatures than CTS below 5 m, SAT and GST are warmer. This might be the result of the interaction between enhanced radiative warming and low ventilation in JJA due to the fact that RPI is surrounded by the canopy of a pine tree forest.

Temperature anomalies with respect to the long-term mean at each depth are shown in Fig. 6 as an example for CTS and HRR. This allows for a more clear visualization of the amplitude attenuation and phase shift with depth. Two cycles are clearly identified again at both sites: the daily cycle, which is visible at GST and hardly noticeable at 0.5 m (Fig. 6a and b), and the annual cycle, with an amplitude reduction and phase shift that are traceable down to 15 m. Nevertheless, there are two differences that are worth to be mentioned. On the one hand, GST and subsurface temperatures underneath at CTS show some constant signal time periods in winter that are not found in SAT, as shown in Fig. 6c, where SAT and GST are represented for a shared period in January 2019 at CTS and HRR. As soon as a snow layer emerges at CTS on January 19, 2019, GST oscillations are supressed despite SAT still varying. This is due to the combination of the soil insulating effect of snow cover (Bartlett et al., 2005) and the zero-curtain effect, which halts temperatures at the ground surface to drop much below zero due to latent heat release (Outcalt et al., 1990). This behaviour is not shared by HRR, since at this lower altitude a regular seasonal



snow layer is less frequent. Hence, HRR is snow-free during the same dates, so SAT and GST variations are strongly coupled. On the other hand, note that SAT variability differs from one site to the other, being the daily (annual) cycle stronger (weaker) at CTS than HRR, as illustrated by a more prominent amplitude (greater differences) of SAT variations in July 2019 with respect

to those in January 2019. This is consistent with SAT annual (intra-annual) variability decreasing (increasing) with altitude in the same area, as shown by Vegas-Cañas et al. (2020).

Air-ground temperature seasonal decoupling is further explored in Table 4, where SAT-GST differences and Pearson's correlation coefficient were computed after filtering out the annual cycle. SAT-GST differences show that on average GST is warmer than SAT, with annual differences being negative. Correlations using daily data are all significant ($p < 0.05$, Student's t-test)

and above 0.58 for all sites. Differences tend to be larger during DJF, likely due to the insulating effect of snow cover, which hampers soil cooling and disrupts SAT-GST coupling. During JJA, significantly warmer GSTs are found at HRR, NVC, and SGV, likely due to surface radiative warming. Additionally, correlation analysis yields the weakest values in DJF at the highest altitude sites, i.e. RPI, HYS, and CTS, which may be linked to the presence of a more steady seasonal snow layer than at the lowest sites. Low correlation is found in JJA at HRR, which might be due to weaker latent heat fluxes connected to Iberian

soils summer drought (Vicente-Serrano et al., 2013). Nevertheless, the diverse local responses at the various sites could be due to land cover differences, water or latent heat exchanges, or other factors (Cermak and Bodri, 2016; Cermak et al., 2017) that have not been further examined herein.

### 4.2   Subsurface thermal regime: the CA

The subsurface thermal structure in the area of the Sierra de Guadarrama is assessed hereafter. The conductive propagation of

GST changes is first addressed by applying a CA (Section 3). Results for CA over the whole profile for every site are included in Fig. 7. As it can be seen CTS, HYS, RPI, and HRR integrate data coming from both BRH 2 m, BRH 20 m, and TRCH 1 m down to 20 m, while NVC and SGV provide TRCH 1 m data. Hence, apparent thermal diffusivity values only account for the first meter underground at these two sites. For the rest of the sites all the information at each location down to 20 m is used to derive apparent thermal diffusivisity estimates. Table 5 segregates results for the case in which only TRCH 1 m would be

considered, as well as for the cases that include information available down to 1 m (TRCH 1 m + BRH 2 m) and down to 20 m using only BRH 20 m. This information complements the estimates provided in Fig. 7.

Thermal diffusivity estimates, i.e. apparent thermal diffusivities, yielded from the slope of the regression lines for both phase shift and amplitude attenuation are very similar for all deep sites, generally in the range of 1-1.3 $10^{-6}$ m$^2$s$^{-1}$. These values are close to the mineral diffusivity values for quartz (ca. 1.4 $10^{-6}$ m$^2$s$^{-1}$) and feldspar (ca. 1.1 $10^{-6}$ m$^2$s$^{-1}$), two of the main

components of gneiss and granite (de Vries, 1963), which are the dominating materials found at the sites of this study (Fig. 2). The observations at NVC and SGV render values out of this range, with apparent diffusivity values between 0.5 and 1.7 $10^{-6}$ m$^2$s$^{-1}$. The shallower observations refer within this first meter of depth to soil material and sediment present in TRCH 1 m rather than the gneiss (granite) characteristic of the other sites at depth. Near the surface the different composition, compactness and soil moisture content can account for the different apparent diffusivity values. In fact, if apparent diffusivity is estimated

at all sites using only TRCH 1 m observations, values in the range of 0.46 (RPI) to 0.88 $10^{-6}$ m$^2$s$^{-1}$ (CTS) are attained





(Table 5). Furthermore, if both TRCH 1 m and BRH 2 m temperatures within the first meter are considered, values hardly change and range from 0.42 (RPI) to 0.87 $10^{-6}$ $m^2s^{-1}$ (CTS). It is often the case that apparent diffusivity values rendered by amplitude attenuation and phase shift within the first meter of the soil can differ considerably, as shown for NVC in Fig. 7. This is an indication that changes in apparent thermal diffusivity occur towards the surface, as seen at NVC and SGV in Fig. 7.

Additionally, changes in the conductive regime may occur within these depths due to soil moisture variability (Melo-Aguilar et al., 2022), freezing and thawing processes (Gao et al., 2008), and near-surface evaporation and latent heat exchanges (Tong et al., 2017) affecting the soil.

The regression spread when considering deep sites in Fig. 7 can be remarkable, scaling up to 25 % at RPI, for instance. Even if the logarithm of the amplitude damping and phase shift for the whole soil column can be considered linear in a first

approach, values do not seem to distribute randomly around the regression line. For instance, between 2 and 7 m all sites tend to show larger amplitude attenuations and phase shifts, hence indicating changes in the regression line slope, i.e. diffusivity, with depth. A further analysis using a two-phase regression model (Fig. 8 Solow, 1987, 1995), confirms there is a tilt in the slope, indicating a significant change in the apparent thermal diffusivity values with depth. Even though the coefficient of determination is very high and significant for every site both for the simple CA in Fig. 7 and for the two-phase CA in Fig.

8 regression lines, always above 0.9, there is a significant reduction of the square sum of residuals in the second case. For instance, this quantity plunges from 5.67 to 1.02 at HYS and from 8.22 to 0.62 at RPI for amplitude attenuation and from 3.21 to 0.91 $day^2$ and 11.46 to 1.20 $day^2$ for phase shift respectively, thus indicating the two-phase regression represents the data more accurately than a single slope linear regression model. Since amplitude attenuation (phase shift) values at every depth were normalised before applying the CA (see Section 3), the change point position is certainly not produced by gathering

information from TRCH 1 m and the BRHs. A change in apparent thermal diffusivity might be indicative of the soil-bedrock transition, where porous non-consolidated is progressively substituted by consolidated and more comprised material, which yields higher diffusivity values with depth (de Vries, 1963). Apparent diffusivity values scale up from 0.7-1 to 1.4-2.6 $10^{-6}$ $m^2s^{-1}$ at most of the sites, which is consistent with the aforementioned transition. However, apparent diffusivity values after the change point might be overestimated due to the small number of points in the linear fit. In Fig. 8, CTS and HYS show

similar change point values for the amplitude attenuation and the phase shift analysis around 5 m. However, there is more uncertainty in the change point at RPI and HRR, where amplitude attenuation yields shallower change points (7 and 4 m) than phase shift analysis (11 and 8 m, respectively).

The previous soil-bedrock transition inferred by the CA can also be identified in Fig. 2 (see Section 2), where the profiles of subsurface composition have a superficial soil layer which is succeeded by accumulated sediments and crystalline materials of

granite and gneiss. Hence, according to Fig. 2, soil-bedrock transition would occur approximately at 8, 6, 5, and 2 m at CTS, HYS, RPI, and HRR respectively, which coincides fairly well with two-phase CA results in Fig. 8 at HYS and RPI, although it overestimates the depth at CTS and underestimates it at HRR.

The afore-explained sample analysis illustrates that changes in subsurface material do not occur abruptly but rather progressively into depth, contributing to changes in apparent thermal diffusivity with depth as well. For that reason, subsurface

diffusivity profiles were derived applying the CA level by level for CTS, HYS, RPI, and HRR (Fig. 9). This shows that diffu-



sivity increases with depth at every site. Whilst RPI and HRR manifest a more linear behaviour, apparent thermal diffusivity values at CTS and HYS differ considerably from one layer to the other, with the profiles being particularly variable near the surface. In general, mean diffusivity values and spread attained in Fig. 7 are well represented by the profiles, being lower (higher) at the shallowest (deepest) layers. Since diffusivity estimates in Fig. 9 are weighted by the layer thickness, small values in the

thinner soil layers near the surface are downweighted and the total vertical estimates tend to be greater than those provided in Fig. 7. Overall, Fig. 9 agrees well with results derived from Figs. 2, 7, 8 and Table 5 and illustrates better the deviations from the linear behaviour for the natural logarithm of amplitude attenuation and the phase shift with depth identified in Fig. 7. These deviations are explained by an increase in apparent thermal diffusivity with depth.

### 4.3 Subsurface thermal regime: the SM

Heterogeneity near the surface and soil moisture spatial and temporal variability can produce changes in diffusivity. To study this effect, the SM is introduced in the analysis, taking into account the damping of all temperature perturbation amplitudes in the frequency spectrum, although overseeing alterations of the conductive regime due to convection, advection or latent heat exchange processes (Gao et al., 2008; Tong et al., 2017). Nevertheless, it can provide some hints on the diffusivity changes with depth and mostly with time. An example of its performance is given in Fig. 10, where it was applied to monthly temperature

data at SGV and NVC in the 1999-2008 period. Apparent thermal diffusivity was retrieved at 0.05, 0.1, 0.25, and 0.5 m layers assessing the spectral amplitude attenuation of GST in its propagation to the 0.1, 0.2, 0.5, and 1 m levels, respectively. Results for both sites show soil thermal diffusivity increases with depth, being higher for NVC than for SGV. The value of apparent thermal diffusivity for the whole soil column derived from the SM at SGV (see points in Fig. 10) agrees with the CA estimate at the annual frequency from Fig. 7 (1/12 month$^{-1}$; 0.55 for CA vs. 0.56 $10^{-6}$ m$^2$s$^{-1}$ for SM), while it is slightly smaller at

NVC (0.82 vs. 0.76 $10^{-6}$ m$^2$s$^{-1}$). The consistency of CA and SM in reproducing the amplitude attenuation of the annual cycle indicates the robustness of SM in reproducing CA results. Furthermore, SM allows for extending the amplitude attenuation analysis to the whole harmonic spectrum. This entails that shorter periods, i.e. higher frequencies than the annual cycle, can be considered. In fact, the application of the SM is not limited to data at monthly resolution, but is applicable to higher temporal resolutions.

Fig. 11 expands the use of the SM as an example to estimate thermal diffusivity values for weekly (a), 3-daily (b), daily (c), and 3-hourly (d) resolutions at HRR. It is observed how the SM consistently reproduces the amplitude attenuation of the annual cycle stemming from CA at every resolution. Moreover, Fig. 11 shows the effect of frequency in amplitude decay: whilst in Fig. 11a, the observed amplitude attenuation curves show an exponential decay in every case, with values over $1/e^2$ except for 1 m, all levels in Fig. 11d are affected by poor signal-to-noise ratios at the high frequencies. Therefore, filtering

out unreliable amplitude attenuation values at high frequencies with a suitable criterium ($e^2$-fold decay in our case) becomes crucial to prevent unbiased thermal diffusivity estimates. If the whole frequency band was considered in the SM, unbiased estimates would still be attained at shallow (low) soil layers (frequencies), but poor performance should be expected at the deep (high) ones.



As it has already been mentioned, soil properties are especially hetereogeneous near the surface, due to changes in soil
material and texture (Cermak et al., 2017) and hydrological (i.e. soil moisture content) changes (Gao et al., 2008; Tong et al.,
2017). The last factor mostly affects subsurface thermal conductivity and volumetric heat capacity by filling up the empty pore
spaces embedded in the soil (Arkhangelskaya and Lukyashchenko, 2018), which in turn modifies apparent thermal diffusivity.
Fig. 11e explores the relation between soil moisture content and apparent thermal diffusivity near the surface at HRR.

The SM at 3-hourly resolution (Fig. 11e) in a 10-day running window is applied to derive a synoptical evolution of apparent
thermal diffusivity at the shallowest layers (0.1 m and 0.2 m), which is subsequently compared to synoptical changes in soil
moisture and precipitation. It is observed how soil moisture presents a seasonal cycle, with extremely low content values in
summer and relatively high but variable values in winter and spring. Transitions between these two states are remarkably abrupt,
with soil moisture content increasing (decreasing) substantially by the beginning (end) of autumn (spring) and responding to
enhanced (supressed) precipitation. On the contrary to other sites in the Sierra de Guadarrama, HRR is rarely snow-covered
in winter, which usually occurs after high-impact snow events (e.g. Filomena, which hit Iberian Peninsula in January 2021,
shading in Figs. 6c, 11e;  Smart, 2021; Tapiador et al., 2021). Apparent thermal diffusivity also shows a seasonal behaviour,
which corresponds to soil moisture low-frequency variability. In summer, poor soil moisture content drives apparent thermal
diffusivity values down. However, even though diffusivity in winter increases with soil moisture content, the relation between
these two variables at the high frequencies is not conclusive. For instance, the evolution of diffusivity in winter-spring 2018 at
both soil levels correctly mimics soil moisture content variations (Fig. 11e), yet fails to capture its strong variations in winter
2020. This agrees on a clear increase of apparent thermal diffusivity from dry to wet soils, but a rather stable diffusivity in
wet soils close to its field capacity observed by Arkhangelskaya and Lukyashchenko (2018). An analogous result for thermal
conductivity by Dai et al. (2019) complies with this result as well. Moreover, SM fails to provide reliable estimates of apparent
thermal diffusivity when the ground is insulated by the snow, due to signal loss in the whole frequency band. This is clearly
shown in Fig. 11e in January 2021, when the R coefficient, that is the square root of the goodness-of-fit of the amplitude
attenuation to an exponential function, becomes extremely poor.

Inter-annual correlation is hard to establish from our assessment due to the shortness of the time series in Fig. 11. However,
previous studies (e.g., Melo-Aguilar et al., 2022) suggest soil moisture and apparent thermal diffusivity changes are propor-
tionally related to some degree. Nevertheless, the lack of a strong evidence of a linear relationship between apparent thermal
diffusivity and soil moisture near the surface could also suggest that heat conduction is no longer the most important mecha-
nism of heat transfer. Heat convective transport via soil moisture and latent energy fluxes, water advection, and horizontal heat
transport due to uneven temperature changes at the ground surface can also contribute to modifying the vertical heat conduction
regime near the surface. The use of a running window SM over longer time series or developing an alternate SM based on a
conductive-convective amplitude attenuation (Gao et al., 2008; Tong et al., 2017) solution to the heat diffusion equation might
better define this relation in wet soils.



## 5  Conclusions

The subsurface thermal structure at six sites in the Sierra de Guadarrama correctly complies with the hypothesis of conduction being the main heat propagation mechanism of SAT changes into depth. On the one hand, SAT changes are transferred to the near-surface at the ground. Although SAT and GST are coupled at long-time scales, snow insulation and zero-curtain effects produce a decoupling in winter in the uppermost stations (CTS, HYS and RPI), which are affected by the presence of a cuasi-steady seasonal snow cap (Figs. 5 and 6). This result agrees with previous studies carried out at cold and snowy sites, such as Fargo (North Dakota, US; Smerdon et al., 2003), Emigrant Pass (Utah, US; Bartlett et al., 2006), Bistriţa (Romania; Demetrescu et al., 2007), and Hveravellir (Iceland; Petersen, 2022).

On the other hand, GST changes are propagated into depth by heat conduction to an extent controlled by the apparent thermal diffusivity of the subsurface. A CA method based on the amplitude attenuation and phase shift of the annual cycle yielded values that are between 1 and 1.3 $10^{-6}$ m$^2$s$^{-1}$ for the whole soil column (i.e. 20 m) at most of the sites (Fig. 7), which are in the range of typical diffusivity values for granite or gneiss, main materials found in the Sierra de Guadarrama's subsurface (Fig. 2). These values are higher than those observed at other sites (e.g. Fargo, Prague, Cape Henlopen; Smerdon et al., 2004), which include shallower measures. If apparent thermal diffusivity is assessed near the surface, the values are smaller and closer to the previous literature (0.4 to 0.8 $10^{-6}$ m$^2$s$^{-1}$; Table 5). A subsequent two-phase adjustment (Fig. 8) and a profile derivation based on CA (Fig. 9) portrays an increase of thermal diffusivity with depth at all sites, being more remarkable near the soil-bedrock transition. Evaluating thermal diffusivity changes with depth using a CA can therefore be a useful approach to determine soil depth at observational sites when no borehole rock samples are available. The deep monitoring profiles at the Sierra de Guadarrama are unique and provide a wider insight into subsurface thermal structure, which may have implications for calibrating soil and bedrock properties in models (Tong et al., 2016; Dai et al., 2019) and monitoring land heat uptake in the terrestrial energy budget (Cuesta-Valero et al., 2022a, b; von Schuckmann et al., 2022).

Near-surface (i.e. 1 m) soil apparent thermal diffusivity vertical and temporal hetereogeneity is further explored by introducing a new SM, which extends the analysis of temperature amplitude attenuation to the whole frequency spectrum. The SM reports consistent apparent thermal diffusivity values with those previously derived with the CA in the first soil meter regardless of the time resolution of the input data at SGV, NVC and HRR. Then, it was used in a synoptical (10-daily) sliding window to explore apparent thermal diffusivity changes in time at the ground surface at HRR (Fig. 11e). Results suggest this parameter presents a seasonal cycle that is positively correlated with soil moisture, mostly under soil drought conditions. However, the fact that these changes are poorly correlated under high soil moisture contents might indicate that heat conduction is not the main heat transfer mechanism in wet soils. Heat transport via water advection and phase changes might not be negligible near the surface, so thermal structure in the shallow soil could be better characterised using a modification of SM that complies with a conduction-convection heat transfer equation (Gao et al., 2008; Tong et al., 2017). Even so, the results show the potential of the SM for inferring soil moisture content changes from soil temperatures, becoming a powerful tool for evaluating soil drought and water resource availability from observational and simulation-based subsurface temperature data.



*Code and data availability.* Quality controlled temperature data at daily resolution and the most relevant codes for data processing used
in this work are available at https://doi.org/10.5281/zenodo.7499419. Nevertheless, all data can be freely obtained for research from the
original data sources, GuMNet and AEMET (see Section 2 for web addresses). Further details of the code are available upon request to the
corresponding author.

*Author contributions.* FGP and JFGR conceptualised and drafted this manuscript. TS, JFGR, and CVC collected the data. FGP processed
the data and developed the methodologies. CVC and CMA contributed to data processing. NJS, PRG, FJCV, AGG, HB, and PdV revised the
manuscript and contributed to the writing and discussion. JFGR supervised this work.

*Competing interests.* The authors declare that they have no conflict of interest.

*Acknowledgements.* This work has been developed within the frame of the GreatModelS (RTI2018-102305-B-C21) and SMILEME (PID2021-
126696OB-C21) projects from the Spanish Ministry of Science and Innovation (MICINN) and CSIC Interdisciplinary Thematic Platform
(PTI) Polar zone observatory (PTI-POLARCSIC). FGP was funded by contract PRE2019-090694 of the MICINN. FJCV was funded by the
Alexander von Humboldt Foundation. We are also thankful for the data provided by the GuMNet (https://www.ucm.es/gumnet/) and AEMET
(https://www.aemet.es/en/datos_abiertos/) meteorological networks.



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





**Table 1.** Name, code, coordinates (longitude, latitude), altitude and timespan of available subsurface temperature data at both the GuMNet and AEMET observational sites.

| Name | Code | Lon. (°) | Lat. (°) | Alt. (m.a.s.l.) | Timespan |
|---|---|---|---|---|---|
| | | | **GuMNet** | | |
| Herrería | HRR | -4.136 | 40.582 | 920 | 2016.06.11 to 2021.03.31 |
| Raso del Pino I | RPI | -3.969 | 40.874 | 1803 | 2017.07.19 to 2020.01.11 |
| Cotos | CTS | -3.961 | 40.825 | 1873 | 2015.09.16 to 2021.03.31 |
| Hoyas | HYS | -3.955 | 40.834 | 2019 | 2015.10.03 to 2021.03.31 |
| | | | **AEMET** | | |
| Segovia | SGV | -4.118 | 40.945 | 1005 | 1989.01.01 to 2012.07.31 |
| P. de Navacerrada | NVC | -4.011 | 40.793 | 1894 | 1998.06.01 to 2018.12.31 |

**Table 2.** Depths corresponding to the available individual temperature series in the trenches and boreholes at each of the sites used in this work (see Table 1 for codes). Dash indicates missing data due to sensor malfunction.

| Soil temperature sensor depths (m) | | | | | | | | | | | | | |
|---|---|---|---|---|---|---|---|---|---|---|---|---|---|
| **TRCH 1 m** | | | | | | **BRH 2 m** | | | | **BRH 20 m** | | | |
| CTS | HYS | RPI | HRR | NVC | SGV | CTS | HYS | RPI | HRR | CTS | HYS | RPI | HRR |
| 0.10 | 0.20 | 0.20 | 0.20 | 0.05 | 0.05 | GST | GST | GST | GST | 1.50 | 1.50 | 1.50 | 1.50 |
| 0.65 | 0.50 | 0.45 | 0.50 | 0.10 | 0.10 | 0.05 | - | 0.05 | - | 2.00 | 2.00 | 2.00 | 2.00 |
| 1.00 | 0.75 | 0.80 | 1.00 | 0.20 | 0.20 | 0.10 | - | 0.10 | 0.10 | 3.00 | - | 3.00 | 3.00 |
| | | | | 0.50 | 0.50 | 0.20 | 0.20 | - | 0.20 | 5.00 | 5.00 | 5.00 | 5.00 |
| | | | | 1.00 | 1.00 | 0.50 | 0.50 | 0.50 | 0.50 | 7.50 | - | 7.50 | 7.50 |
| | | | | | | 1.00 | 1.00 | 1.00 | 1.00 | 10.00 | 10.00 | 10.00 | 10.00 |
| | | | | | | 1.50 | 1.50 | 1.50 | 1.50 | - | 15.00 | 15.00 | 15.00 |
| | | | | | | 2.00 | 2.00 | 2.00 | 2.00 | 20.00 | 20.00 | 20.00 | - |





**Figure 1.** (a) Subsurface data availability over the area of the Sierra de Guadarrama used in this work. The map in the inset (left upper corner) points out the location of the Sierra de Guadarrama in the Iberian Peninsula. The extension covered by the map in (a) is represented by the orange shaded area in the inset. (Right) Conceptual sketches of subsurface equipment for 1 m trenches (b), 2 m (c) and 20 m boreholes (d). The same distribution of subsurface temperature (ST) sensor depths is carried out at every site (c, d). Sensors in trenches are unevenly placed within the first meter of the ground at different sites. Pale yellow (grey) background color refers to soil (bedrock) in all panels. The transition depth between soil and bedrock (blurred area in d) is only intended for illustration and takes place at different depths for each site. Colors and shapes of the symbols are also illustrative. In addition to subsurface temperature data, all the stations provide SAT measurements as well as soil moisture content (SH) in the trenches.





**Figure 2.** Subsurface mineral composition of the borehole cores extracted when drilling at CTS, HYS, RPI, and HRR. The top soil layer (red) covers approximately the first soil meter at every site, whilst sediments (brown) depth varies from one site to the other. Bedrock (magenta) underneath is of gneiss or granite, normally weathered at the top. In the case of CTS, some photos of the material at different depths are also included for illustration.



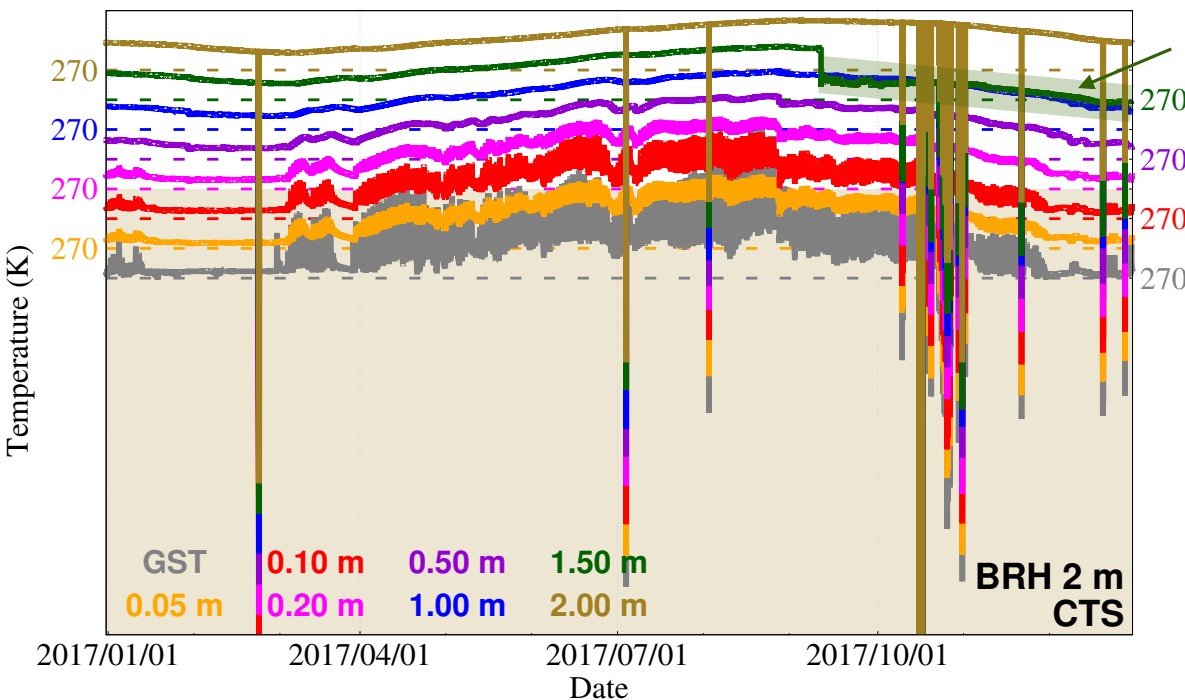

**Figure 3.** Example of errors in BRH 2 m at CTS in year 2017. The 10-minute GST observations (grey) and those of temperatures for seven levels below the surface are shown. Time series are shifted an offset equivalent to 10 K for clearer visualization. The value of 270 K is shown for all levels as a reference. The olive color band at the bottom represents the level of 230 K for the 2 m depth temperature time series, used to determine nonphysical values. The shaded area for the level of 1.5 m pointed by the arrow indicates a time interval eliminated due to a shift in the mean temperature value.

**Table 3.** Record of periods discarded at CTS and HYS for the analysis subsequently performed in this work. Records are sorted by log depths (descending) and span from the dates indicated up to the present.

| Code | Sensor | Depth (m) | Timespan (from) |
|------|--------|-----------|-----------------|
| CTS | BRH 2 m | 1.5 | 2017.09.11 |
| | BRH 20 m | 3.0 | 2019.05.21 |
| | BRH 20 m | 2.0 | 2019.04.09 |
| | BRH 20 m | 1.5 | 2018.08.19 |
| HYS | BRH 2 m | 2.0 | 2018.08.31 |
| | BRH 2 m | 1.5 | 2018.09.14 |
| | BRH 2 m | 1.0 | 2017.09.21 |
| | BRH 20 m | 2.0 | 2019.11.03 |





**Figure 4.** SAT, GST, and subsurface temperatures (K) at HRR. Subsurface temperatures stem from the BRH 20 m (black lines), BRH 2 m (brown lines), and TRCH 1 m (orange lines). Blue dashed lines show the resulting fit to an annual cycle at each depth.



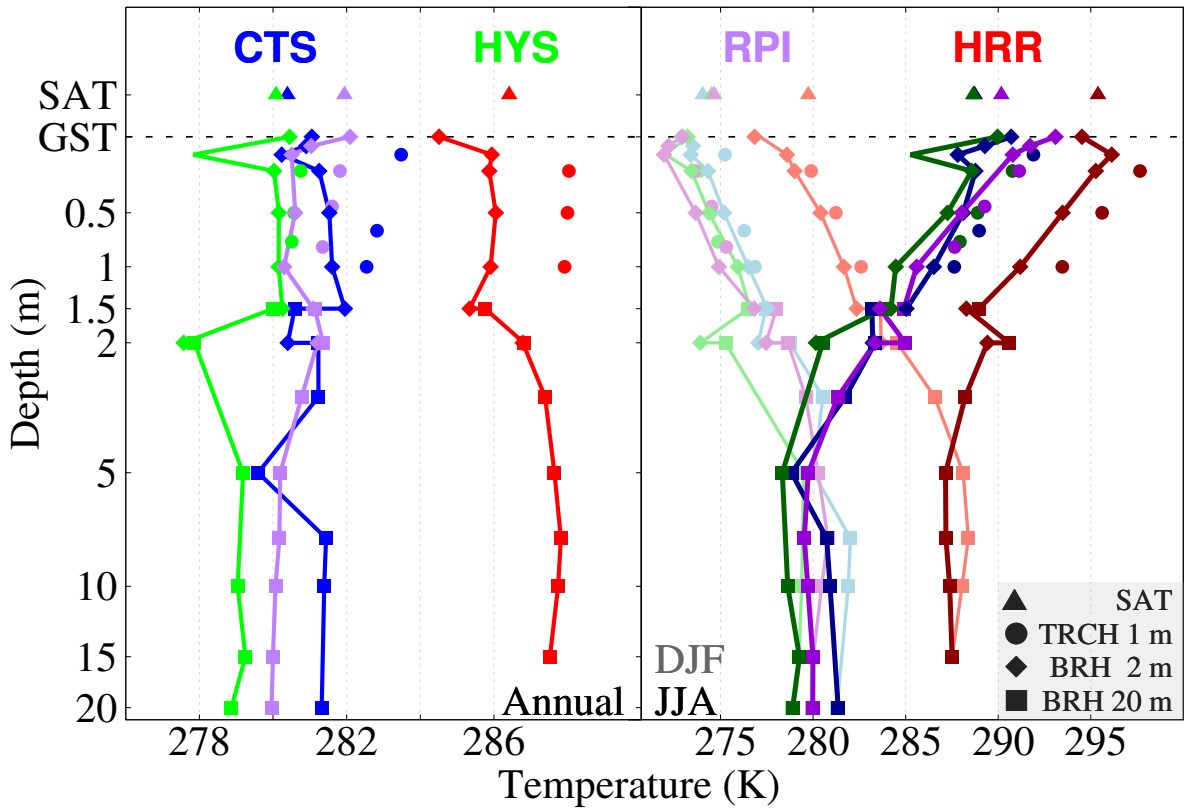

**Figure 5.** SAT (triangles) and subsurface temperature profiles (diamonds for BRH 2 m, squares for BRH 20 m, and circles for TRCH 1 m) at CTS (blue), HYS (green), RPI (purple), and HRR (red) for their respective whole time spans (see Table 1). Annual mean temperatures are shown on the left hand side of the figure, whilst JJA (DJF) are shown on the right in dark (light) colors.

**Table 4.** SAT-GST differences in mean (K) and Pearson's correlation coefficient using daily data. Annual wave was previously filtered out both from SAT and GST data for this analysis. Non-significant ($p < 0.05$) differences and correlations are shown in bold.

| Code | SAT - GST (K) | | | Correlation coefficient | | |
|------|--------|-------|-------|--------|------|------|
| | Annual | JJA | DJF | Annual | JJA | DJF |
| CTS | -1.80 | **0.21** | -3.45 | 0.61 | 0.57 | 0.54 |
| HYS | -0.47 | 0.52 | -0.59 | 0.68 | 0.83 | 0.43 |
| RPI | -0.92 | **-0.34** | -2.37 | 0.58 | 0.75 | **0.16** |
| HRR | -1.58 | -2.36 | **-0.11** | 0.59 | 0.44 | 0.74 |
| NVC | -1.99 | -3.52 | -1.38 | 0.71 | 0.56 | 0.67 |
| SGV | -0.46 | -1.65 | **0.20** | 0.81 | 0.76 | 0.77 |






**Figure 6.** SAT, GST, and subsurface temperature anomalies (K) with respect to the annual mean at CTS (a) and HRR (b). (c) SAT (light) and GST (°C; dark colors) changes during 10 days in January (blue) and July (red) 2019 and evolution of the snow cover (m; light pink) at CTS and HRR.





**Figure 7.** Phase shift in days (blue) and logarithm of amplitude ratios (red) vs. depth at CTS, HYS, RPI, HRR, NVC, and SGV (see Table 1). Points in orange, brown and black stem from a least-squares adjustment for sinusoidal fitting of the annual cycle of TRCH 1 m, BRH 2 m, and BRH 20 m data, respectively. Diffusivity values ($10^{-6}$ m$^2$s$^{-1}$) are retrieved by linearly fitting all the points available at every site (see text for details). Regression lines, confidence intervals using a significance level of $p = 0.05$ and diffusivity values for the logarithm of the amplitude ratios (phase shift) are included in red (blue).





**Table 5.** Apparent thermal diffusivity values ($10^{-6}$ $m^2s^{-1}$) obtained from the amplitude attenuation, ln(A/A$_0$), and phase shift, $\phi$, for TRCH 1 m only (first column), the first meter of depth using a blend of TRCH 1 m and BRH 2 m data (second column), and BRH 20 m (third column) at CTS, HYS, RPI, HRR, SGV, and NVC.

| | **TRCH 1 m** | | **1 m (TRCH 1 m + BRH 2 m)** | | **BRH 20 m** | |
|---|---|---|---|---|---|---|
| | $ln(A/A_0)$ | $\phi$ | $ln(A/A_0)$ | $\phi$ | $ln(A/A_0)$ | $\phi$ |
| **CTS** | $0.88 \pm 0.16$ | $0.85 \pm 0.06$ | $0.87 \pm 0.57$ | $0.70 \pm 0.26$ | $1.41 \pm 0.21$ | $1.44 \pm 0.38$ |
| **HYS** | $0.73 \pm 0.28$ | $1.26 \pm 0.33$ | $0.45 \pm 0.15$ | $0.58 \pm 0.33$ | $1.38 \pm 0.29$ | $1.07 \pm 0.13$ |
| **RPI** | $0.46 \pm 0.18$ | $0.84 \pm 0.29$ | $0.42 \pm 0.09$ | $0.67 \pm 0.20$ | $1.41 \pm 0.27$ | $1.49 \pm 0.40$ |
| **HRR** | $0.50 \pm 0.17$ | $0.69 \pm 0.21$ | $0.45 \pm 0.12$ | $0.68 \pm 0.10$ | $1.20 \pm 0.16$ | $1.15 \pm 0.19$ |
| **NVC** | $0.82 \pm 0.51$ | $1.75 \pm 1.24$ | | | | |
| **SGV** | $0.55 \pm 0.24$ | $0.77 \pm 0.07$ | | | | |



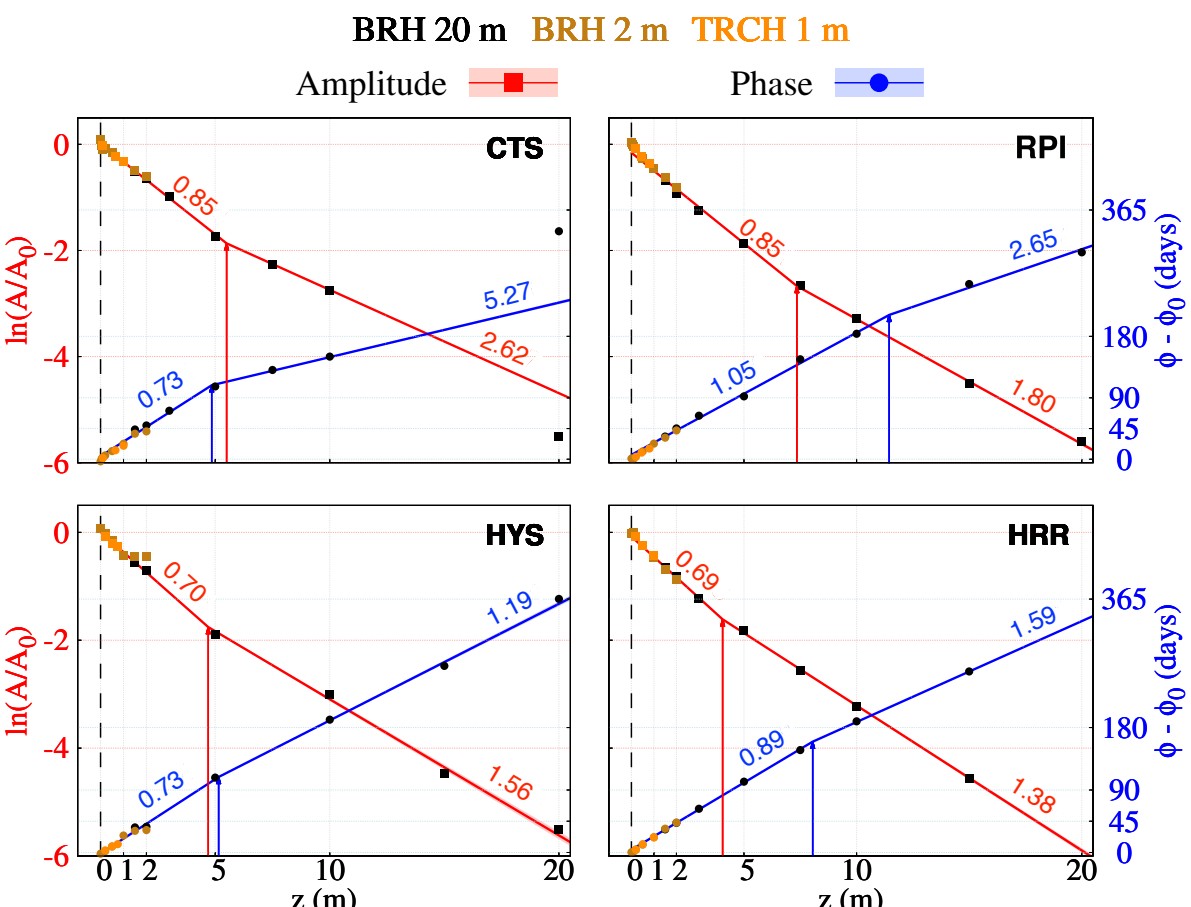

**Figure 8.** Two-phase regression of the points shown in Fig. 7. Lines represent the linear trends that are the best fit to the data before and after the estimated point of change for the logartithm of the amplitude ratio (phase shift), depicted in red (blue). Apparent thermal diffusivity values ($10^{-6}$ m$^2$s$^{-1}$) before and after the change point are given as well in every case. The 20 m level at CTS was omitted from the analysis.

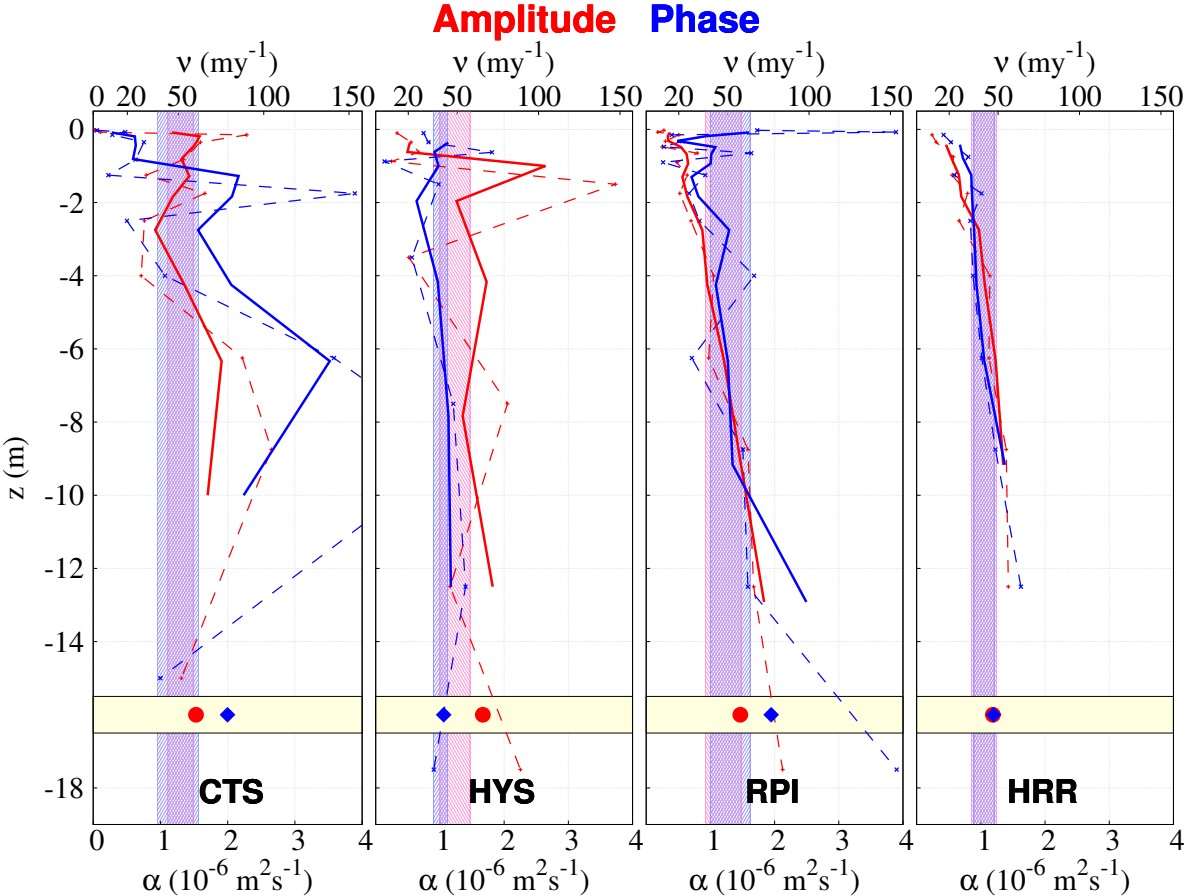

**Figure 9.** Thermal diffusivity ($\alpha$, $10^{-6}$ $m^2 s^{-1}$) and wave velocity ($\nu$, m $yr^{-1}$) profiles with depth (m). The red curves were obtained from normalised amplitude attenuation values $ln(A/A_0)$ and the blue ones from phase shift values with depth. Dashed (solid) lines represent the raw (3-running layer thickness-weighted average) estimate. Points at the bottom of the profile (shaded yellow) represent the thickness-weighted average of the whole subsurface profile. Shaded areas in red (blue) represent the thermal diffusivity values obtained by linearly fitting amplitude attenuation (phase shift) in Fig. 7.



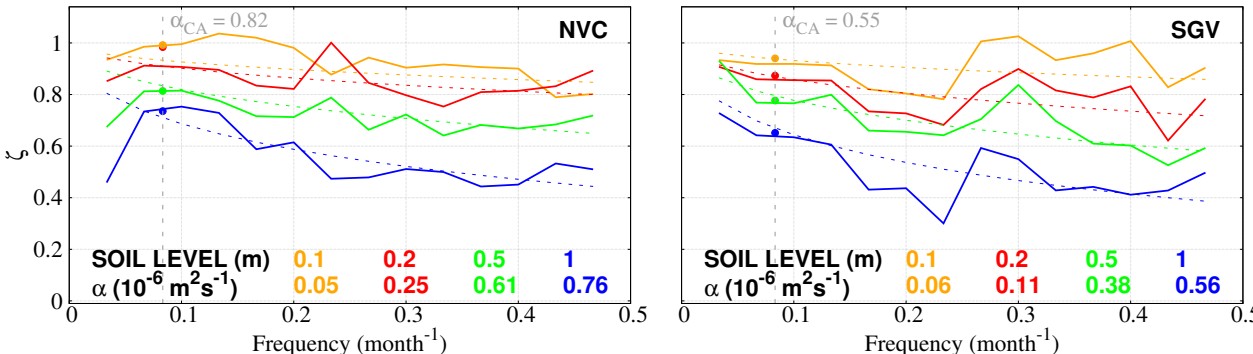

**Figure 10.** SM amplitude attenuation analysis at NVC (left panel) and SGV (right panel). Monthly time series at both sites were selected for a common interval spanning from 1999 to 2008. Missing data were linearly interpolated. Spectral attenuation curves are calculated as a ratio of amplitudes of the Welch's periodogram at (Stoica and Moses, 1997) 0.1 (orange), 0.2 (red), 0.5 (green), and 1 (blue) m temperature levels with respect to GST. Solid (dashed) lines represent observed (theoretical) curves at every level. Diffusivity values below come from least-squares fitting the observed to theoretical curves. Diffusivity values in grey (top left of each panel) show the result of the CA for the amplitude attenuation of the annual cycle (1/12 month$^{-1}$) shown in Fig. 7. Colored points at the annual frequency (vertical grey dashed line) show the corresponding result of the CA for the amplitude attenuation at every level with respect to GST, which can be compared with the corresponding SM observed (solid) and theoretical (dashed) values.



**Figure 11.** As in Fig. 10, SM analysis of subsurface temperature attenuation at HRR for weekly (a), 3-daily (b), daily (c), and 3-hourly (d) data. The hollow parts of the observed attenuation curves were excluded in the estimation of apparent thermal diffusivity, according to the $e^2$-fold decay criterium explained in Section 3. (e) Precipitation, soil moisture content and estimated near-surface thermal diffusivity from applying the SM at the 0.1 m (light orange) and 0.2 m (light red) levels with respect to the GST with a 10-day running window at 3-hourly resolution; 11-day low pass running averages are provided in dark colors. Soil moisture content at 4 cm is shown at daily (10-daily) resolution in light blue (blue). Precipitation at 10-daily resolution is shown in purple. Least-square regression correlation (R) for SM at every time step is given below in each case. The gray shaded period represents the time interval with snow insulation in Fig. 6c for HRR.