# Peer review of "Thermodynamic and hydrological drivers of the soil and bedrock thermal regimes in Central Spain"

_EGUsphere, 2023_

## Referee Comment (RC2)

[referee-annotated manuscript omitted]

---

## Author Response (AR1)

Félix García-Pereira[1], Jesús Fidel González-Rouco[1], Thomas Schmid[2], Camilo Melo-Aguilar[3], Cristina Vegas-Cañas[1], Norman Julius Steinert[4], Pedro José Roldán-Gómez[1], Francisco José Cuesta-Valero[5], Almudena García-García[5], Hugo Beltrami[6], and Philipp de Vrese[7]

[1]Complutense University of Madrid, Faculty of Physical Sciences, and Geosciences Institute (UCM-CSIC), Madrid, Spain
[2]Department of Environment, CIEMAT, Madrid, Spain
[3]Balearic Ocean Centre, Spanish Institute of Oceanography (IEO-CSIC), Palma de Mallorca, Spain
[4]NORCE, Norwegian Research Centre, Climate and Environment, Bergen, Norway
[5]Helmholtz-Centre for Environmental Research (UFZ), Leipzig, Germany
[6]Climate and Atmospheric Sciences Institute, St. Francis Xavier University, Antigonish, Canada
[7]Max Planck Institute for Meteorology, Hamburg, Germany

**Correspondence:** Félix García-Pereira (felgar03@ucm.es), Cristina Vegas-Cañas (cvegas@ucm.es)

The authors would like to thank the reviewers for their constructive suggestions and the time they devoted to reading and proofreading the manuscript. We have tried to integrate all suggestions and think that the manuscript has improved with them. We do appreciate their contribution.

The next sections contain a detailed point-by-point response to the reviewers' comments. Comments are labeled by reviewers and in order of appearance, i.e. R2C3 is the third comment of reviewer 2. The original number by the reviewer is also preserved if it was given.

**1 Anonymous Referee 1**

*GENERAL COMMENTS:*

R1C0: *REVIEWER'S COMMENT:*

*This is an interesting study that presents a kind of dataset that is rare, namely continuous measurements of subsurface temperatures over multiple depths exending meters to tens of meters into the ground. The authors do a good job of contextualizing their work in the broader literature and provide some insightful analyses of the data. I provide a detailed list of revisions below, but my general assessment is that the paper should be published after my comments are addressed. While none of my comments require substantial changes, the number of small changes that are necessary amount to a necessary major revision.*

AUTHORS' RESPONSE:

**The authors welcome the positive perspective of the reviewer on the paper. We are grateful for the reviewer's**

**comments.**

**Please find below the comprehensive point-to-point response to your review.**

R1C1: *REVIEWER'S COMMENT:*

*Ln 3: two shallow "profiles"*

AUTHORS' RESPONSE:

**The text has been changed according to the reviewer's indication.**

R1C2: *REVIEWER'S COMMENT:*

*Ln 13: thermal diffusivity, not heat*

AUTHORS' ANSWER:

**The text has been changed according to the reviewer's indication.**

R1C3: *REVIEWER'S COMMENT:*

*Ln 17: "and ongoing anthropogenic"*

AUTHORS' ANSWER:

**The text has been changed according to the reviewer's indication.**

R1C4: *REVIEWER'S COMMENT:*

*Ln 20: has allowed "the attribution of".*

AUTHORS' ANSWER:

**The text has been changed according to the reviewer's indication.**

R1C5: *REVIEWER'S COMMENT:*

*Ln 23: The difference between land and ocean warming rates is not exclusively due to evaporative cooling. The heat capacities are important, as is the circulation of the ocean. An increase in air temperature is also not the principal means by which the ocean is warming, but rather changes in the energy balance at the land and ocean surfaces that is driving both air and ocean temperature change.*

AUTHORS' ANSWER:

**The text has been changed following the reviewer's indications to: "At the global scale, the increase of air temperature affects the ocean and land surface disproportionally**

, with warming in the oceans being smaller mostly due to its larger heat capacity. Hence, while oceans have warmed about 0.88 $^{\circ}C$ since the last half of the 19th century, land has experienced a temperature increase of 1.59 $^{\circ}C$ (Chen et al., 2021). It is important to understand how this temperature rise has propagated into the soil and subsoil, which has been shown to be important for land surface processes and meteorological extreme events (Miralles et al., 2019; Zhou et al., 2019; Seneviratne et al., 2021)Miralles et al. (2019) and Zhou et al. (2019) are new references." (see lines 26 in the annotated manuscript).

R1C6: *REVIEWER'S COMMENT:*

*Ln 25: Rahmstorf and Coumou is a strange reference here. There are many other papers on land-atmosphere interactions that would be more appropriate.*

AUTHORS' ANSWER:

**This reference has been substituted following this indication, including a couple of papers by Miralles et al. (2019), and Zhou et al. (2019) (see line 31 in the annotated manuscript).**

R1C7: *REVIEWER'S COMMENT:*

*Ln 27: long-term scales should be changed to be specific (over decades to centuries). Melo-Aguilar et al. as a singular reference is also an insufficient reference for this statement. There is a long history of comparisons between GST reconstructions and SAT comparisons that make this case (see the work of Harris and Chapman, Huang and Pollack, Beltrami, etc.).*

AUTHORS' ANSWER:

**The text has been changed following the indication of the sentence to: "Ground surface temperature (GST) and surface air temperature (SAT) are coupled at multi-decadal and centennial scales (Pollack and Huang, 2000; Beltrami and Kellman, 2003; Melo-Aguilar et al., 2018)Pollack and Huang (2000) and Beltrami et al. (2003) are new references, and GST perturbations are subsequently propagated into depth by heat conduction (Carslaw and Jaeger, 1959)" (see line 33 in the annotated manuscript). Moreover, references to Pollack and Huang (2000) and Beltrami and Kellman (2003) have been included as additional support to this statement, as indicated by the reviewer.**

R1C8: *REVIEWER'S COMMENT:*

*Ln 29: SAT changes are not transmitted into the subsurface. As stated above, the temperature changes in both SAT and GST are the product of changes in the energy balance at the land surface.*

AUTHORS' ANSWER:

**As this statement might be misleading, it has been removed following the reviewer's indication.**

**R1C9:** *REVIEWER'S COMMENT:*

*Ln 36: The polar changes do not have implications exclusively for climate in high latitude regions. For instance, emissions of carbon from permafrost has global implications.*

AUTHORS' ANSWER:

**The paragraph has been changed as a response to R2C4, and now potential impact of polar changes in global climate is mentioned in line 49: "Moreover, soil warming has relevant impacts on high latitude regions, where permafrost thawing can produce changes in soil hydrology (Andresen et al., 2020; Burke et al., 2020) and eventually subsurface collapse (Abbott and Jones, 2015; Pegoraro et al., 2021), posing risks to infrastructures (Rotta Loria, 2023)-Rotta Loria (2023) is a new reference- and reshaping landscapes. Moreover, permafrost thawing triggers the release of pollutants (e.g., Schaefer et al., 2020) and massive amounts of carbon (Turetsky et al., 2019) to the atmosphere, potentially affecting global climate (e.g., de Vrese et al., 2023)".**

**R1C10:** *REVIEWER'S COMMENT:*

*Ln 38: temperatures allow recovery of past SAT*

AUTHORS' ANSWER:

**The text has been changed according to the reviewer's indication.**

**R1C11:** *REVIEWER'S COMMENT:*

*Ln 43: sources is crucial. It is also not obvious what "its" is refering to is in this sentence. SAT-GST coupling? If so, SAT-GST isn't such an important thing to understand in the context of climate change, rather the relationship is important for understanding other crucial processes.*

AUTHORS' ANSWER:

**"Its" is referring to both processes, temperature coupling at the ground surface and heat conductive propagation in depth. However, it is true that this sentence can be interpreted as SAT-GST being a key process to understanding climate change, as the reviewer remarks. Moreover, this manuscript does not focus on assessing SAT-GST coupling and heat conduction at multi-decadal or multi-centennial time scales, but on understanding how these two processes operate locally from short observational time series. For these two reasons, we have decided to remove this sentence from the text (see line 55 in the annotated manuscript).**

**R1C12:** *REVIEWER'S COMMENT:*

*Ln 45: The list of references here is pretty limited. Other important papers include the following:*
*Baker, D. G. and D. L. Ruschy: The recent warming in eastern Minnesota shown by ground temperatures, Geophys. Res. Lett., 20(5), 371–374, doi:10.1029/92GL02724, 1993.*
*Putnam, S. N. and Chapman, D. S.: A geothermal climate change observatory: first year results from Emigrant Pass*

*in northwest Utah, J. Geophys. Res., 101, 21877–21890, 1996.*

*Smerdon, J. E., Pollack, H. N., Enz, J. W., and Lewis, M. J.: Conduction-dominated heat transport of the annual temperature signal in soil, J. Geophys. Res., 108, 2431, doi:10.1029/2002JB002351, 2003.*

*Smerdon, J. E., Pollack, H. N., Cermak, V., Enz, J. W., Kresl, M., Safanda, J., and Wehmiller, J. F.: Air-ground temperature coupling and subsurface propagation of annual temperature signals, J. Geophys. Res., 109, D21107, doi:10.1029/2004JD005056, 2004.*

*Smerdon, J. E., Pollack, H. N., Cermak, V., Enz, J. W., Kresl, M., Safanda, J., and Wehmiller, J. F.: Daily, seasonal, and annual relationships between air and subsurface temperatures, J. Geophys. Res., 111, D07101, doi:10.1029/2004JD005578, 2006.*

*Smerdon, J. E., Beltrami, H., Creelman, C., and Stevens, M. B.: Characterizing land-surface processes: A quantitative analysis using air-ground thermal orbits, J. Geophys. Res., 14, D15102, doi:10.1029/2009JD011768, 2009.*

AUTHORS' ANSWER:

**The references to Putnam and Chapman (1996); Smerdon et al. (2004, 2006) have been included following the reviewer's indication (see line 59 in the annotated manuscript). However, to keep the number of references to well-known processes moderate, following the R2C9 reviewer's indication, we decided not to include the citation to Baker et al. (1993), which is more focused on temperature trends analysis, Smerdon et al. (2003), as it somewhat superseded by Smerdon et al. (2004). A citation to Smerdon et al. (2009) has been included as part of the evidence of SAT-GST coupling and conductive propagation from the modeling perspective (see line 62 in the annotated manuscript).**

R1C13: *REVIEWER'S COMMENT:*

*Ln 53: This discussion is missing an acknowledgment that trends have also been investigated (not exclusively a harmonic assessment) in Lesperance et al. (2010). The harmonics have also been tracked on an annual basis in thermal orbit applications, i.e. an approach that could theoretically be done for a single year of data (Sushama et al. 2007 and Smerdon et al. 2009)*

AUTHORS' ANSWER:

**The text has been changed following the reviewer's indication to mention the study of the conductive propagation of long-term temperature trends with depth and the annual wave propagation analysis by means of thermal orbits. It has also been indicated that these two studies assume a priori constant values of thermal diffusivity, while in this work no a priori assumption is made and this parameter is determined as a way to characterize differences in the subsurface thermal structure between sites. The modified text is:** "Previous works assess the propagation of temperature trends (Lesperance et al., 2010)(Lesperance et al., 2010 is a new reference), and thermal orbits (Smerdon et al., 2008), using

a priori defined constant values of thermal diffusivity. Other studies tackled the propagation of single frequency harmonic signals with depth, i.e. the annual cycle (Hurley and Wiltshire, 1993; Smerdon et al., 2003) , in order to derive estimates of the apparent thermal diffusivity. Nevertheless [...]" (see line 68 in the annotated manuscript).

R1C14: *REVIEWER'S COMMENT:*

*Ln 56: subsurface temperature for at least*

AUTHORS' ANSWER:

**The text has been changed according to the reviewer's indication.**

R1C15: *REVIEWER'S COMMENT:*

*Ln 64: contribute to more accurate*

AUTHORS' ANSWER:

**The text has been changed according to the reviewer's indication.**

R1C16: *REVIEWER'S COMMENT:*

*Ln 65: The ending clause of this sentence is confusing. I think the authors are making a reference to the amount of heat that has been stored in the terrestrial subsurface over the last 50-100 years, relative to other components of the climate system. That should be clarified. They should also cite Beltrami et al. (2002) in the context of their assessment.*

AUTHORS' ANSWER:

**The text has been changed following the reviewer's indication: "A better characterization of subsurface thermal properties can also contribute to  more accurate land heat uptake estimates, i.e. energy storage in the subsurface due to industrial warming, which is the second component contributing the most to terrestrial energy partitioning (Beltrami et al., 2002; Cuesta-Valero et al., 2022b; von Schuckmann et al., 2022)Beltrami et al. (2002) is a new reference" (see line 80 in the annotated manuscript).**

R1C17: *REVIEWER'S COMMENT:*

*Ln 66: compliance is not the right word in this sentence*

AUTHORS' ANSWER:

**The text has been changed following the reviewer's indication to: "This paper assesses the validity of the..." (see line 84 in the annotated manuscript).**

R1C18: *REVIEWER'S COMMENT:*

*Ln 73: This allows us to test*

AUTHORS' ANSWER:

**The text has been changed according to the reviewer's indication.**

R1C19: *REVIEWER'S COMMENT:*

*Ln 78: "multi-annual long temporal intervals" is overburdened. Revise for brevity and clarity.*

AUTHORS' ANSWER:

**The word "long" is redundant and thus removed from the sentence, so it is clearer now.**

R1C20: *REVIEWER'S COMMENT:*

*Ln 83: also allows diffusivity changes to be evaluated through time and connected to*

AUTHORS' ANSWER:

**The text has been changed according to the reviewer's indication.**

R1C21: *REVIEWER'S COMMENT:*

*Ln 85: This form of finger counting the following sections is unneccessary. Outlining the questions addressed in the manuscript is useful at this stage, but enumerating the sections is not. Either reframe as questions or remove this part of the intro.*

AUTHORS' ANSWER:

**Since the key points and questions of the paper are enumerated in the last paragraphs of the introduction, this paragraph was removed according to the reviewer's indication.**

R1C22: *REVIEWER'S COMMENT:*

*Ln 89: mountan system that splits*

AUTHORS' ANSWER:

**The text has been changed according to the reviewer's indication.**

R1C23: *REVIEWER'S COMMENT:*

*Ln 133: Note the decrease in high-frequency variability with subsurface depth, which is consistent with*

AUTHORS' ANSWER:

**The text has been changed according to the reviewer's indication.**

R1C24: *REVIEWER'S COMMENT:*

*Ln 142: The authors detail the method of subsurface temperarture collection in great detail, but say nothing of how the SAT and snow depth measurements are collected. A little detail is warranted here. It should also be specifically mentioned at what height the SAT is measured. Vegetation cover at the sites would also be worth noting. If the authors have a picture of a representative sight, that would also be useful.*

AUTHORS' ANSWER:

**Regarding how SAT is monitored, a sentence has been included following the reviewer's comment: "SAT is recorded at 2 m above the ground surface at all sites using a temperature probe encapsulated within a shield" (see line 170 in the annotated manuscript). Concerning snow depth collection, some additional information has also been included: "Snow depth is measured with an ultrasonic snow depth sensor that captures the distance to the surface. Depth values are obtained from the subtraction of surface level measured from the reference ground surface level value" (line 171). Finally, a short description of the vegetation cover in the area where sites are settled has been included at the beginning of the section: "In all cases, vegetation cover at the sites consists of small grass that can show minimal changes during the year. No site has trees or shrubs within the fenced perimeter of the monitoring station" (line 119). After careful consideration, we have decided not to include pictures of the sites since the vegetation and layout are quite similar between them and would make the manuscript, already quite extensive, even larger.**

R1C25: *REVIEWER'S COMMENT:*

*Ln 155: Smerdon and Steiglitz (2006) also discuss the harmonic solution in this context and provide the explicit expressions for the amplitude and phase evolution as a function of depth.*

AUTHORS' ANSWER:

**The reference to Smerdon and Stieglitz (2006) was included following the reviewer's indication (see line 184 in the annotated manuscript).**

R1C26: *REVIEWER'S COMMENT:*

*Ln 166: The authors use this parenthetical convention at multiple places within the manuscript. It is a poor and confusing form that should be avoided. This is not just my own hobby horse - there have been papers in our field about it:*
*Robock, A. (2010), Parentheses Are (Are Not) for References and Clarification (Saving Space), Eos Trans. AGU, 91( 45), 419– 419, doi:10.1029/2010EO450004.*

AUTHORS' ANSWER:

**The parenthetical convention used across the text has been corrected following the reviewer's indication. Some examples can be found in lines 189, 191, 195, 252, 259, and 273.**

R1C27: *REVIEWER'S COMMENT:*

*Ln 176: each harmonic is also propagated*

AUTHORS' ANSWER:

**The text has been changed according to the reviewer's indication.**

R1C28: *REVIEWER'S COMMENT:*

*Ln 183: The conductive process is widely described as a lowpass filter in this literature, but the description is not accurate and should be avoided. Higher frequencies are indeed damped more strongly with depth than lower freqencies, but as the authors have noted, each frequency is also phase shifted relative to each other. This latter property is not characteristic of a lowpass filter, which would not shift the relative phases of the signal. While this may seem pedantic, it has important implications and the community should be careful to avoid this misleading description.*

AUTHORS' ANSWER:

**The statement "the subsurface acts as a lowpass filter" has been removed according to the reviewer's indication (see line 217 in the annotated manuscript).**

R1C29: *REVIEWER'S COMMENT:*

*Ln 200: A few points about the spectral method. It does not appear to me that it is any different than the CA, but merely exploits multiple frequencies outside of the annual signal. This is novel and useful, but it does not warrent a billing as a different approach, i.e. it is merely exploiting the properties of conduction across a wider frequency band than a method that exclusively tracks the annual signal. On a very small note, it might be worth renaming the approach to yield an acronym that is not SM. This is a widely used acronym for soil moisture, which makes it a bit confusing when used later in the manuscript.*

AUTHORS' ANSWER:

**We acknowledge the caveats raised by the reviewer concerning the spectral method introduced by this work. It is true that the spectral method is based on resolving the one-dimensional heat conduction equation, tackling amplitude attenuation with depth as shown by the previous literature. However, doing so by going beyond the single annual cycle harmonic requires the introduction of the spectral amplitude attenuation, and its least-square adjustment to an exponential function, which we think is novel enough to say we are following an "alternative approach" to the thermal diffusivity determination. Regarding the SM acronym, we have changed it to SpM throughout the manuscript to accommodate the reviewer's indication.**

R1C30: *REVIEWER'S COMMENT:*

*Ln 211: deterministic is not the right word here*

AUTHORS' ANSWER:

**The text has been changed following this indication to: "A first inspection reveals that the only reg-ular temperature variation at all levels, both in timing and magnitude, is the annual cycle." (see line 236 in the annotated manuscript).**

R1C31: *REVIEWER'S COMMENT:*

*Ln 214: despite the fact that*

AUTHORS' ANSWER:

**The text has been changed according to the reviewer's indication.**

R1C32: *REVIEWER'S COMMENT:*

*Ln 215: The B.I.G. is notorious, but the damping of the annual signal is not. Consider a different word.*

AUTHORS' ANSWER:

**"Notorious" was replaced by "noteworthy" according to the reviewer's indication.**

R1C33: *REVIEWER'S COMMENT:*

*Ln 217: Further should be Furthermore in several cases within this paragraph.*

AUTHORS' ANSWER:

**The text has been changed according to the reviewer's indication.**

R1C34: *REVIEWER'S COMMENT:*

*Ln 228: slightly weaker is vague. Do you mean the amplitude of SAT is smaller?*

AUTHORS' ANSWER:

**Yes, that was what we meant. The sentence has been corrected according to the reviewer's indication: "With regard to SAT-GST coupling, Fig. 5 shows the annual cycle is less pronounced for SAT than GST due to radiative..." (see line 273 in the annotated manuscript).**

R1C35: *REVIEWER'S COMMENT:*

*Ln 229: "being SAT-GST anomalies negative" is incorrect. Rephrase for clarity.*

AUTHORS' ANSWER:

**The text has been changed following this indication to: "** resulting in a negative SAT-GST offset in summer and positive in winter**" (see line 273 in the annotated manuscript).**

R1C36: *REVIEWER'S COMMENT:*

*Ln 234: The general statement before noting that RPI and CTS do not follow the expected behavior with elevation is that the sites conform to the expectation. But that leaves only two other sites! This should be reframed to reflect the fact that two sites behave as expected and two sites don't.*

AUTHORS' ANSWER:

**The text has been changed following this indication to: "Subsurface temperatures with depth follow the distribution of SAT** at HYS and HRR, with HRR exhibiting higher values for the whole profile. However, this pattern is not observed at RPI and CTS, as RPI shows lower temperatures below 5 m despite SAT and GST being warmer. **This might be the result of the interaction between enhanced radiative warming and low ventilation in JJA due to the fact that RPI is surrounded by the canopy of a pine tree forest." (see line 278 in the annotated manuscript).**

R1C37: *REVIEWER'S COMMENT:*

*Ln 241: that are worth mentioning*

AUTHORS' ANSWER:

**The text has been changed according to the reviewer's indication.**

R1C38: *REVIEWER'S COMMENT:*

*Ln 245: There are many studies that have investigated the impact of snow on GST. In addition to some that have already been mentioned, there are also these:*
*Stieglitz, M., S. J. Dery, V. E. Romanovsky, and T. E. Osterkamp (2003), The role of snow cover in the warming of arctic permafrost, Geophys. Res. Lett., 30(13), 1721, doi:10.1029/2003GL017337.*
*Zhang, T. (2005), Influence of the seasonal snow cover on the ground thermal regime: An overview, Rev. Geophys., 43, RG4002, doi:10.1029/2004RG000157.*

AUTHORS' ANSWER:

**The reference to Zhang (2005) was included following the reviewer's indication (see line 292 in the annotated manuscript). However, this is not a review article and to keep the number of references to well-known processes**

moderate, following the R2C9 reviewer's indication, we decided not to include a citation to Stieglitz et al. (2003), which is more focused on the impact of snow cover on the thermal offset at the regional scale of the Arctic.

R1C39: *REVIEWER'S COMMENT:*

*Ln 300: This is also discussed in the following:*

*Pollack, H. N., J. E. Smerdon, and P. E. van Keken (2005), Variable seasonal coupling between air and ground temperatures: A simple representation in terms of subsurface thermal diffusivity, Geophys. Res. Lett., 32, L15405, doi:10.1029/2005GL023869.*

AUTHORS' ANSWER:

**The reference to Pollack et al. (2005) was included following the reviewer's indication (see line 350 in the annotated manuscript).**

R1C40: *REVIEWER'S COMMENT:*

*Ln 310: The authors presumably have the stratigraphy of the soil and bedrock horizons from the cores that the drilled. Why not confirm whether these estimated transitions align with their observations?*

AUTHORS' ANSWER:

**The paragraph has been extended in order to make a clearer comparison between the results coming from the stratigraphy and the CA soil-bedrock detection, according to the reviewer's suggestion: "The previous** changes in apparent diffusivity **inferred by the CA can**  **be** partially traced to the soil-bedrock transitions observed **in Fig. 2 (see Section 2), where the profiles of subsurface composition have a superficial soil layer which is succeeded by accumulated sediments and crystalline materials of granite and gneiss.**  Results coming from the CA and the stratigraphy of the samples agree at HYS, placing the soil-bedrock transition at a depth of 5 m. However, if the change points should be due to the bedrock limit, the CA underestimates the observed depth at CTS, where the bedrock starts in the sampled cores at 8 m (Fig. 8). The fact that the amplitude attenuation and phase shift CA results do not coincide at RPI and HRR might be also an indication that other factors influence the occurrence of change points (e.g. soil moisture), the CA failing again to produce consistent amplitude and phase estimates for change points, both above the value of sampled cores that show the soil-bedrock transition at 5 and 2 m, respectively. Considering the combined results from the stratigraphy and the CA for the four sites, it becomes clear that there is some uncertainty in the relationship between subsurface material and apparent thermal diffusivity changes, and other influences play a role in producing the step changes in thermal properties with depth." (see line 356 in the annotated manuscript).**

R1C41: *REVIEWER'S COMMENT:*

*Ln 315: This diffusivity estimate as a function of depth was done in Smerdon et al. (2003).*

AUTHORS' ANSWER:

**The reference to Smerdon et al. (2003) was included following the reviewer's indication (see line 373 in the annotated manuscript): "This shows that diffusivity increases with depth at every site, which agrees with a previous analysis conducted at Fargo (Smerdon et al., 2003)".**

R1C42: *REVIEWER'S COMMENT:*

*Ln 345: criterion*

AUTHORS' ANSWER:

**The text has been changed according to the reviewer's indication.**

R1C43: *REVIEWER'S COMMENT:*

*Ln 350: Again, this variable diffusivity due to surface processes was also discussed by Pollack et al. (2006).*

AUTHORS' ANSWER:

**The reference to Pollack et al. (2005) was included following the reviewer's indication (see line 409 in the annotated manuscript).**

R1C44: *REVIEWER'S COMMENT:*

*Ln 359: In constrast to tother sites*

AUTHORS' ANSWER:

**The text has been changed according to the reviewer's indication.**

R1C45: *REVIEWER'S COMMENT:*

*Ln 382: is consistent with*

AUTHORS' ANSWER:

**The text has been changed according to the reviewer's indication.**

R1C46: *REVIEWER'S COMMENT:*

*Ln 386: quasi-steady, but it is not clear exactly what is meant here.*

AUTHORS' ANSWER:

**The presence of a snow cap is almost continuous at these three sites in winter, since snow melting is limited by**

the low temperatures. The text has been changed to: "Although SAT and GST are coupled at long-time scales, snow insulation and zero-curtain effects produce a decoupling in winter in the uppermost stations (CTS, HYS, and RPI), which are affected by the presence of a  seasonal snow cap (Figs. 5 and 6)." (see line 446 in the annotated manuscript).

R1C47: *REVIEWER'S COMMENT:*

*Ln 389: propagated to depth*

AUTHORS' ANSWER:

**The text has been changed according to the reviewer's indication.**

R1C48: *REVIEWER'S COMMENT:*

*Ln 393: The construction here implies that the diffusivities at other sites should compare to those determined for the Spanish sites. Why should that be so? It is distinctly possible that the diffusivities are simply reflective of different suburface materials, which is expected.*

AUTHORS' ANSWER:

**The authors wanted to highlight that the values being similar are the ones derived from the CA and SpM for each of the sites independently, that is, the HRR CA value and the HRR SpM value, and the same for SGV and NVC. Following the reviewer's indication, the sentence has been changed to: "The SpM reports consistent apparent thermal diffusivity values in the first soil meter at SGV, NVC, and HRR with those previously derived with the CA for each of the respective sites and regardless of the time resolution of the input data." (see line 463 of the annotated text).**

R1C49: *REVIEWER'S COMMENT:*

*Ln 405: The authors use synoptical in multiple places in the manuscript. The implied meaning is not obvious.*

AUTHORS' ANSWER:

**"Synoptical" was used with the intention of referring to thermal diffusivity and soil moisture changes driven by changes in the synoptic-scale spatial patterns (i.e. succession of extratropical highs and lows, passage of fronts, etc.). As synoptical might not be the most precise term, it has been substituted by "intra-monthly" following the reviewer's indication (see line 467 in the annotated manuscript).**

R1C50: *REVIEWER'S COMMENT:*

*Figure 3: There are no temperature tick marks on this graph to vive a sense of the magnitude of the temperature variation. Only the mean values of offset temperature are marked. I also find the green shading very hard to see*

*and understand what it is marking. Moving the arrow to the beginning of the offest might improve things, as would providing a caption next to the arrow describing what it is pointing at.*

AUTHORS' ANSWER:

**Ticks every 5 degrees have been incorporated following the reviewer's indication. Moreover, the green shading has been removed and the arrow moved to the beginning of the temperature shift at the 1.5 m level. The legend has been changed as well to accommodate the description of the new figure's layout.**

R1C51: *REVIEWER'S COMMENT:*

*Figure 5: There are some strange discontinuities in these figures. For instance, do the authors think the feature between 1.5 and 2 m in the annual CTS BRH profile is real?*

AUTHORS' ANSWER:

**A new version of Fig. 5 has been produced where the profiles of each borehole and the trench are plotted separately. Indeed, there is an offset at the 1.5 and 2 m depth levels when comparing temperatures from the 2 m and 20 m deep boreholes. The authors have cautiously checked if these discontinuities, which are most remarkable at HYS, might be due to differing missing periods. After time masking the subsurface series to have common missing/available data, the conclusion is that is not the cause behind these strange offsets in the mean and the temperature values at each level are indeed different. After analysing again the borehole core samples and the features of the sites, the authors reached the conclusion these differences may be due to heterogeneities in the terrain material and soil moisture, even when borehole installations are separated by only a few tens of centimeters. Anyway, the offsets in Fig. 5 between BRH 2 and BRH 20 m represent differences in the mean state, and they do not affect the subsequent analysis presented in Figs. 7, 8, and 9, since a normalization was done precisely to avoid inhomogeneities from subsurface temperatures coming from different installations. The text has been modified to describe the issue in Fig. 5: "JJA and DJF temperature profiles converge at a depth of around 5 m at all sites to then cross over, which is a sign of the phase shift of the annual cycle. The transition from the BRH 2 m to the BRH 20 m profiles show noticeable discontinuities that are of the order of 2.7 K and 2.2 K at HYS and about 1.5 K and 1 K at HRR for the 1.5 and 2 m temperature levels, respectively. These discontinuities are most likely due to heterogeneous soil and sediment formations. The uppermost soil and sediment layers vary according to the sites and in all cases reach a depth greater than 2 m (Fig. 2). In the case of HYS, for instance, the greatest annual temperature difference is observed between the 2 m and 20 m boreholes at 1.5 m depth. The site is on a middle slope with a slope gradient of 10 to 15% and the soil horizons have a sandy loam to a sandy clay loam texture. What is most notable is the varying content of rock fragments (15 to >80%) within the different soil horizons. This is further observed throughout the sediment layer where rock fragment distribution varies with depth and is very weathered. Therefore, temperature variations over short distances and within the uppermost soil and**

**sediment layers are expected due to their different properties which will also influence the water infiltration and circulation.** (see line 262 in the annotated manuscript).

R1C52: *REVIEWER'S COMMENT:*

*Figure 6: This figure would be a lot easier to read with a legend instead of a color description in the caption.*

AUTHORS' ANSWER:

**A legend has been incorporated in Figure 6c so SAT and GST in summer and also snow in winter can be easily identified, following the reviewer's suggestion.**

R1C53: *REVIEWER'S COMMENT:*

*Figure 11e: This figure would be a lot easier to read with a legend instead of a color description in the caption. The grey shaded area also looks more like a line, making the description of that feature in the graph confusing.*

AUTHORS' ANSWER:

**All the information concerning the different time series gathered in panel 11e is either indicated in the y-axis when there is one single depth level (precipitation, soil moisture content), or in orange/red when it represents 0.1/0.2 m soil levels (thermal diffusivity and regression correlation). The only information that is not explicitly shown in the figure is light colors referring to bulk data, while dark ones to 11-day running averages. Since the panel is already quite loaded and after carefully considering the reviewer's suggestion, we have decided not to modify the figure. We did modify the caption to better describe the grey area, as follows: "The grey shaded thick line represents the only time interval (4 days)...".**

**2 Anonymous Referee 2**

*GENERAL COMMENTS:*

R2C0: *REVIEWER'S COMMENT:*

*This is a very interesting study presenting data that are rare, linking shallow surface and deeper soil temperature regimes over time. Near to the end of the Abstract the novelty is clear, but this is lost to some extent in the main body of the paper, where much of the presentation is limited to the data, rather than presenting what is new and exciting in the work. There is emphasis placed on well-described phenomena, such as near surface temperature variability due to the drivers of soil formation, but less on the interesting aspects to link the shallow surface to deeper soil.*

AUTHORS' RESPONSE:

**The authors acknowledge the good perspective of the reviewer on the paper. Please find below the comprehensive point-to-point response to your review.**

*SPECIFIC COMMENTS:*

R2C1: *REVIEWER'S COMMENT:*

*Abbreviations are used throughout. Some of these are not necessary and they make following the paper difficult.*

AUTHORS' ANSWER:

**The authors have cautiously revised the manuscript to seek for redundant or unnecessary abbreviations and acronyms. Unfortunately, all of the acronyms are extensively and repeatedly (more than 20 times each) used throughout the manuscript, as is the case for the station codes (see Table 1), the names of the different setups (BRH, TRCH, see Figure 1) and variables (SAT, GST, ST), or the names of the seasons (JJA, and DJF). The soil moisture, SH, and the subsurface temperature, ST, acronyms are used only a few times and were removed from the text. If there are some other abbreviations the reviewer considers may be avoided and suppressed, we will remove them from the text.**

R2C2: *REVIEWER'S COMMENT:*

*The title would be better and more novel if instead of 'subsurface' it mentioned 'from surface soil to bedrock'.*

AUTHORS' ANSWER:

**After careful consideration, we have decided to change the manuscript's title to "Thermodynamic and hydrological drivers of the soil and bedrock thermal regimes in Central Spain" following the reviewer's indication.**

R2C3: *REVIEWER'S COMMENT:*

*The Abstract is good but try to bring some of the novelty presented at the end earlier on in the text. This will convince readers that the paper is new and worth reading. The early part of the abstract reads like an observational study, but your paper is more than this.*

AUTHORS' ANSWER:

**The Abstract was modified following the reviewer's indication to present the novel SpM methodology introduced by this work earlier on in the text. Please see the annotations made across the abstract in the track changes document attached to the revision.**

R2C4: *REVIEWER'S COMMENT:*

*The Introduction needs some restructuring. The text varies from assuming no knowledge to having expert knowledge, and the flow of arguments could be clearer. In one sentence you mention permafrost, then many other parameters, and then permafrost again. Make it easier for the reader to follow.*

AUTHORS' ANSWER:

**The Introduction has been modified to make the rationale easier to be followed, according to the reviewer's indication. For instance, a piece of text which better illustrates the disproportional warming of ocean and land due to climate change (see the response to comment R1C5 of the revision) or an improved framing of this work's aim within the previous literature (see the response to comment R1C13 of the revision) have been included. Moreover, the paragraph tackling the effects of land warming in permafrost regions has been rephrased for better understanding, according to the reviewer's indications (see line 45 in the annotated manuscript).**

R2C5: *REVIEWER'S COMMENT:*

*You need to be clearer on the research gap before describing your study. Remind readers of the 'conductive hypothesis'. Missing is some of the novelty. The coupling of shallow to deep thermal regimes is novel, so bring this out more in the last paragraphs.*

AUTHORS' ANSWER:

**A better explanation of the research gap was addressed as a response to R1C13. This study is relatively novel in assessing heat conduction from a batch of deep continuous observational records coming from different sites, so a sentence was included to highlight it in the second-to-last paragraph of the introduction (see line 88 in the annotated manuscript). It also introduces a new methodology or methodological variant, the SpM, to overcome the timescale constraint of the annual wave, but the novelty of this is already mentioned in the abstract (response to comment R2C3) and in the last paragraph of the introduction. A comment clarifying the conductive hypothesis was also introduced according to the reviewer's comment (see line 54 in the annotated manuscript), and method-**

ologies tackling it in previous literature were extensively explained as a response to comment R1C13 (see line 68 in the annotated manuscript).

R2C6: *REVIEWER'S COMMENT:*

*Section 2 contains most of the necessary information and indicates an interesting study. Where you describe the sensors at line 100 it would help to be clearer about depth intervals. You miss describing the date range of the dataset.*

AUTHORS' ANSWER:

**Depth intervals at which sensors are placed are described in Table 2. As it was perhaps not clear in the paragraph, we added a couple of sentences at the end to indicate it: "Table 2 provides a detailed list of the depths at which subsurface temperature sensors were installed. Levels with malfunctioning sensors are marked with a dash" (see line 132 of the annotated manuscript). Date ranges are included in Table 1, and a clarifying sentence commenting on it was also added at the beginning of the paragraph: "Table 1 includes a more detailed description of the name, code, geographical position, and date range of available data of each location" (see line 110 of the annotated manuscript).**

R2C7: *REVIEWER'S COMMENT:*

*Section 3 – give an indication of the extension of time in the text at line 148. The text is clear to follow.*

AUTHORS' ANSWER:

**We realized the meaning of "the assessment is extended in time..." is not clear. The text has been changed to clarify this: "The SAT-GST seasonal coupling is explored by quantifying the offset and correlation of seasonal (December-January-February, DJF; June-July-August, JJA) and annual 2 m air with GSTs taken from the top level  temperatures in the BRH 2 m set up. The estimates are obtained from the complete time span of available data at each site (Tables 1,2)." (see line 176 of the annotated manuscript).**

R2C8: *REVIEWER'S COMMENT:*

*Section 4 – please see comments on the marked version.*

AUTHORS' ANSWER:

**Most of the changes indicated in the document attached to the revision have been incorporated. The changes have been marked in the document with track changes.**

R2C9: *REVIEWER'S COMMENT:*

*You spend too much time writing about well known processes, such as shallow surface temperature variation, and not enough linking shallow and deeper soils.*

AUTHORS' ANSWER:

**The focus of this paper is on applying the well-known heat diffusion analysis to characterize the subsurface (soil and bedrock) thermal structure for a batch of observational records, using first the annual wave and afterward a spectral adaptation of the method to the whole harmonic range. We hope this is more clear now from the revised versions of the Abstract, Introduction, and Methodology parts. Please see the responses to comments R1C13, R1C29, R2C3, R2C4, and R2C5.**

[revised manuscript text omitted]

---

## Author Response (AR2)

Félix García-Pereira1, Jesús Fidel González-Rouco1, Thomas Schmid2, Camilo Melo-Aguilar3, Cristina Vegas-Cañas1, Norman Julius Steinert4, Pedro José Roldán-Gómez1, Francisco José Cuesta-Valero5, Almudena García-García5, Hugo Beltrami6, and Philipp de Vrese7 1Complutense University of Madrid, Faculty of Physical Sciences, and Geosciences Institute (UCM-CSIC), Madrid, Spain 2Department of Environment, CIEMAT, Madrid, Spain 3Balearic Ocean Centre, Spanish Institute of Oceanography (IEO-CSIC), Palma de Mallorca, Spain 4NORCE, Norwegian Research Centre, Climate and Environment, Bergen, Norway 5Helmholtz-Centre for Environmental Research (UFZ), Leipzig, Germany 6Climate and Atmospheric Sciences Institute, St. Francis Xavier University, Antigonish, Canada 7Max Planck Institute for Meteorology, Hamburg, Germany

Correspondence: Félix García-Pereira (felgar03@ucm.es), Cristina Vegas-Cañas (cvegas@ucm.es)

The authors would like to thank the reviewers for their constructive suggestions and the time they devoted to reading and proofreading the manuscript. We have tried to integrate all suggestions and think that the manuscript has improved with them. We do appreciate their contribution.

The next sections contain a detailed point-by-point response to the reviewers' comments. Comments are labeled by reviewers and in order of appearance, i.e. R2C3 is the third comment of reviewer 2. The original number by the reviewer is also preserved if it was given.

**1 Anonymous Referee 1**

GENERAL COMMENTS:

**R1C0: REVIEWER'S COMMENT:**

General Remarks: The authors have comprehensively responded to the reviewers' comments. This manuscript should be published after the minor revisions described below.

**AUTHORS' RESPONSE:**

The authors welcome the positive perspective of the reviewer on the response to the revision. We are grateful for the reviewer's minor comments. Please find below the comprehensive point-to-point response to your review.

**R1C1: REVIEWER'S COMMENT:**

Ln 15: "The spectral method is capable of"

**AUTHORS' RESPONSE:**

**The text has been changed according to the reviewer's indication.**

**R1C2: REVIEWER'S COMMENT:**

Ln 15: Correlation has a specific statistical meaning and is used here to refer to a vague association between two physical quantities in the preceding sentence. Its use is therefore confusing and therefore should be avoided, while more clearly articulating what relationship the authors are discussing at this point.

**AUTHORS' ANSWER:**

To avoid using the term "correlation", the sentence has been modified following the reviewer's comment to: "Re- sults with the spectral method suggest that changes in near-surface thermal diffusivity are related to changes in soil moisture content changes. The spectral method shows to be capable of detecting this correlation at short time scales, which makes it a potential tool in soil drought and water resource availability reconstruction, which makes it a potential tool to gain information about soil drought and water resource availability from soil temperature data." (see line 14 of the annotated manuscript)

**R1C3: REVIEWER'S COMMENT:**

Ln 23: This is one thing that the authors did not fully improve. The air temperature increases are not the changes in the ocean and land surface warming! The energy imbalance is what causes the changes in air, ocean, and land temperatures and the authors should be more careful about their language here.

**AUTHORS' ANSWER:**

The text has been changed according to the reviewer's indication: At the global scale, the increase of air temperature affects the ocean and land surface disproportionally, with warming in the oceans being smaller mostly due to its larger heat capacity. Hence, while oceans have warmed about  $0.88 \, ^{\circ}C$  since the last half of the 19th century, land has experienced a temperature increase of 1.59  $^{\circ}C$  (Chen et al., 2021) This warming has been larger over land than over the ocean, with temperature values in 2011-2020 being 1.59  $^{\circ}C$  and  $0.88 \, ^{\circ}C$  higher than 1850-1900, respectively (Chen et al., 2021). (see line 25 of the annotated manuscript)

**R1C4: *REVIEWER'S COMMENT*:**

Ln 37: Infrastructure should not be plural.

**AUTHORS' ANSWER:**

The text has been changed according to the reviewer's indication.

**R1C5: REVIEWER'S COMMENT:**

Ln 41: GST not SAT histories

**The text has been changed according to the reviewer's indication.**

**R1C6: REVIEWER'S COMMENT:**

Ln 59: "for at least a few years." Also begin the following sentence with "Such monitoring." "That" is a vague reference.

**AUTHORS' ANSWER:**

**The text has been changed according to the reviewer's indication.**

**R1C7: REVIEWER'S COMMENT:**

Ln 64: "implications for"

**AUTHORS' ANSWER:**

**The text has been changed according to the reviewer's indication.**

**R1C8: REVIEWER'S COMMENT:**

Ln 71: There are multiple places in the manuscript where vague language continues to be used. This is a good example. "Short" and "relatively long" is unhelpful here. Simply indicate explicitly how long each of the records are.

**AUTHORS' ANSWER:**

An indication to the length of the records in each case has been added following the reviewer's indication: This is achieved by determining the apparent soil thermal diffusivity from four short (4-6 years) and two relatively longlonger (ca. 20 years) subsurface temperature records obtained at six sites in the area of the Sierra de Guadarrama (see line 74 of the annotated manuscript).

**R1C9: REVIEWER'S COMMENT:**

Ln 72: What is "its" referring to here?

**AUTHORS' ANSWER:**

"Its" here is referring to subsurface temperature variability. Therefore, "its" has been changed to "temperature" so as to improve the clarity of this sentence, according to the reviewer's indication.

**R1C10: REVIEWER'S COMMENT:**

Ln 73: Unique means singular and does not have qualifiers. Something is either unique or it is not. Avoid quite unique, very unique, etc.

**The text has been changed according to the reviewer's indication.**

**R1C11: REVIEWER'S COMMENT:**

Ln 74: High mountain area relative to what? Again, relatively is vague and has not context here.

**AUTHORS' ANSWER:**

The word "relatively" has been eliminated, according to the reviewer's indication.

**R1C12: REVIEWER'S COMMENT:**

Ln 77: "The propagation of the annual wave with depth is subsequently studied."

**AUTHORS' ANSWER:**

The text has been changed according to the reviewer's indication.

**R1C13: REVIEWER'S COMMENT:**

Ln 92: "hereinafter"

**AUTHORS' ANSWER:**

The text has been changed according to the reviewer's indication.

**R1C14: REVIEWER'S COMMENT:**

Ln 102: "information on"

**AUTHORS' ANSWER:**

**The text has been changed according to the reviewer's indication.**

**R1C15: REVIEWER'S COMMENT:**

Ln 103: "short grass that changes minimally during"

**AUTHORS' ANSWER:**

**The text has been changed according to the reviewer's indication.**

**R1C16: REVIEWER'S COMMENT:**

Ln 133: "easier to detect"

**AUTHORS' ANSWER:**

**R1C17: REVIEWER'S COMMENT:**

Ln 133: "physically implausible"

**AUTHORS' ANSWER:**

The text has been changed according to the reviewer's indication.

**R1C18: REVIEWER'S COMMENT:**

Ln 141: "is consistent with a conductive process?

**AUTHORS' ANSWER:**

The sentence has been modified following the reviewer's indication to: "Note the decrease in high-frequency variability with subsurface depth, which is consistent with conductive lawa heat conduction process (Carslaw and Jaeger, 1959)."

**R1C19: REVIEWER'S COMMENT:**

Ln 144: "of one such"

**AUTHORS' ANSWER:**

**The text has been changed according to the reviewer's indication.**

**R1C20: REVIEWER'S COMMENT:**

Ln 183: "This CA framework"

**AUTHORS' ANSWER:**

**The text has been changed according to the reviewer's indication.**

**R1C21: REVIEWER'S COMMENT:**

Ln 201: "Because the periodogram calculation"

**AUTHORS' ANSWER:**

The text has been changed according to the reviewer's indication.

**R1C22: REVIEWER'S COMMENT:**

Ln 208: "To prevent poor signal-"

**AUTHORS' ANSWER:**

**R1C23: REVIEWER'S COMMENT:**

Ln 209: "from biasing the analysis"

**AUTHORS' ANSWER:**

**The text has been changed according to the reviewer's indication.**

**R1C24: REVIEWER'S COMMENT:**

Ln 214: There appears to be some inconsistent reasoning at this point. It is indicated a few sentences above that the CA method is used to estimate the near-surface soil thermal diffusivity values, which are then used in the SpM framework. It is also argued that the SpM method can be used to analyze shorter records, relative to the CA method. It is therefore not clear how this latter fact can be true if a CA estimate is used in the SpM approach. I trust there is an explanation for this, but the way it is written in the manuscript it gives the impression of a contradiction.

**AUTHORS' ANSWER:**

The SpM approach can be applied regardless of the length, the time frequency or the depth of the subsurface temperature records. However, as it is explained in the methodology section, the spectral attenuation curves are quite noisy at the high frequencies, so the authors included the  $e^2$ -fold decay strategy to cut-off the curve at a certain frequency and filter out these noisy signals. In the manuscript, this cut-off frequency was calculated from thermal diffusivity values coming from the CA, yet the same could be done assuming a realistic value of the soil thermal diffusivity (e.g.  $10^{-6} m^2/s$ ). The following plot (Fig. R1) shows how this would work for three values of thermal diffusivity (0.6. 1, and 1.4  $10^{-6} m^2/s$ ) for three-hourly and daily near-surface temperature data coming from HRR (the same used in Fig. 11 in the manuscript). The thermal diffusivity estimates slightly increase when assuming higher a priori diffusivity values for the cut-off frequency, but changes are very small in comparison to the range of unceirtainty of CA diffusivity estimates within the first meter of the soil (see Table 5). This demonstrates assuming a certain a priori thermal diffusivity for computing the  $e^2$ -fold decay cut-off frequency is perfectly feasible, rendering the SpM-based thermal diffusivity estimates completely independent form CA-based ones. One possibility would be to include in the manuscript the attached graph as a response to the reviewer. However, the authors believe it is more appropriate to keep the current figure in the main document (Fig. 11 a,b,c,d), since this figure better illustrates the consistency between the CA and the SpM. A piece of text was added instead to clarify this aspect in the Methodology section: "To determine this frequency at every site and level, near-surface soil thermal diffusivity values coming from the CA are used. However, it is possible to use a physically plausible prior thermal diffusivity value (ca.  $10^{-6} m^2 s^{-1}$ ) to calculate this frequency, leading to very similar results and thus making the SpM independent from the CA." (see line 220 of the annotated manuscript)